# SPTNet: An Efficient Alternative Framework for Generalized Category Discovery with Spatial Prompt Tuning

**Hongjun Wang**[1]  **Sagar Vaze**[2]  **Kai Han**[1†]

[1]Visual AI Lab, The University of Hong Kong
[2]Visual Geometry Group, University of Oxford
hjwang@connect.hku.hk   sagar@robots.ox.ac.uk   kaihanx@hku.hk

## Abstract

Generalized Category Discovery (GCD) aims to classify unlabelled images from both 'seen' and 'unseen' classes by transferring knowledge from a set of labelled 'seen' class images. A key theme in existing GCD approaches is adapting large-scale pre-trained models for the GCD task. An alternate perspective, however, is to adapt the *data* representation itself for better alignment with the pre-trained model. As such, in this paper, we introduce a two-stage adaptation approach termed SPTNet, which iteratively optimizes *model parameters* (*i.e.*, model-finetuning) and *data parameters* (*i.e.*, prompt learning). Furthermore, we propose a novel spatial prompt tuning method (SPT) which considers the spatial property of image data, enabling the method to better focus on object parts, which can transfer between seen and unseen classes. We thoroughly evaluate our SPTNet on standard benchmarks and demonstrate that our method outperforms existing GCD methods. Notably, we find our method achieves an average accuracy of 61.4% on the SSB, surpassing prior state-of-the-art methods by approximately 10%. The improvement is particularly remarkable as our method yields extra parameters amounting to only 0.117% of those in the backbone architecture. Project page: https://visual-ai.github.io/sptnet.

## 1 Introduction

Deep learning models have been extensively studied in image recognition He et al. (2016); Krizhevsky et al. (2017), typically relying on large-scale annotated data, as well as a 'closed-world' assumption: that the data to be classified shares the same classes as the labelled training data. However, this assumption limits application to real-world scenarios where the target data contains 'unseen' classes images alongside 'seen' ones Han et al. (2019; 2020; 2021); Fini et al. (2021); Wen et al. (2023); Jia et al. (2021); Zhao & Han (2021). Recently, Category Discovery (CD) has emerged as a practical open-world learning problem, where a model trained using partially labelled data is tasked to categorize unlabelled data that may originate from unseen categories. Initially, it was studied as Novel Category Discovery (NCD) Han et al. (2019) focusing on unlabelled data exclusively from unseen categories. Subsequently, it was extended to Generalized Category Discovery (GCD) Vaze et al. (2022) encompassing unlabelled data from both seen and unseen categories.

State-of-the-art GCD methods Vaze et al. (2022); Cao et al. (2022); Wen et al. (2023) employ pre-trained self-supervised models, such as DINO Caron et al. (2021), and partially fine-tune their parameters on the target task, taking advantage of the strong generalization properties of these representations. In this paradigm, data remains fixed while iterating over the model. However, fully fine-tuning a large pre-trained model can lead to overfitting to the labelled data, and is computationally expensive. Instead of focusing solely on the model, we find that alternatives which manipulate the *data* to cater to the model, are both more efficient and can also achieve better GCD performance. Specifically, *visual prompting* methods (*e.g.*, Jia et al. (2022); Bahng et al. (2022)), have recently been explored to improve model capability by modifying the input or intermediate features through the addition of extra learnable tokens. Although these methods are effective in fully supervised learning, they do not improve representations for generalization and struggle to achieve satisfactory performance in the open-world GCD task. A natural approach to integrating both advantages

---

†Corresponding author.

is to simultaneously optimize the model and data parameters. However, this non-convex bilevel optimization often leads to sub-optimal solutions for both sets of parameters.

Inspired by the expectation–maximization (EM) algorithm Dempster et al. (1977) and decomposition techniques for bilevel optimization Engelmann et al. (2020); Byeon & Van Hentenryck (2022), we introduce a two-stage iterative learning framework called SPTNet for GCD, optimizing both model parameters (*i.e.* model-finetuning) and data parameters (*i.e.* prompt learning). More specifically, the framework includes two phases: (1) In the first phase, the backbone model is frozen, and only the prompts are adjusted. (2) In the second phase, we fix the prompt parameters and update the backbone model with a contrastive loss, using an augmented data pair constructed by the raw image together with its prompted version. The prompts and model are alternately trained until convergence. In this way, our learned prompt can be considered as a learned augmentation, targeted for the downstream recognition task (see Fig. 1).

Following arguments in the GCD literature Vaze et al. (2022), that object parts are an effective vehicle to transfer knowledge between 'seen' and 'unseen' categories, we propose Spatial Prompt Tuning (SPT), which learns *pixel-level* prompts around local image regions. Unlike previous methods (*e.g.*, Jia et al. (2022); Bahng et al. (2022)) that introduce learnable tokens to the hidden model space, or wrap prompts around the entire image border, SPT divides the original image into patches and attaches prompts to each patch in pixel space. The objective of SPT is to achieve improved alignment between the large pre-trained model and discriminative image regions in the target task. We conduct experiments on seven datasets using the standard evaluation protocol in the GCD setting. Our method achieves an average accuracy of 61.4%, which is higher than the previous state-of-the-art methods by around 10%, in proportional terms, on the SSB benchmark Vaze et al. (2021). Remarkably, this improvement is achieved by introducing only 0.117% extra parameters compared to all ViT-Base parameters, demonstrating the efficiency and effectiveness of our approach.

Our contributions can be summarized as follows: (1) We introduce a two-stage iterative learning framework called SPTNet, integrating advantages of both model parameters (*i.e.*, model-finetuning) and data parameters (*i.e.*, prompt learning) learning for GCD. (2) We propose a new spatial prompt method (called SPT) to adapt the data representation for better alignment with the pre-trained model. The method learns independent prompts for different spatial regions and introduces only 0.039% additional parameters compared to all ViT-Base parameters. (3) We conduct comprehensive evaluations of our method on seven datasets, including three generic (*i.e.*, CIFAR-10, CIFAR-100, and ImageNet-100) and four fine-grained benchmarks (CUB, Stanford Cars, FGVC-Aircraft, and Herbarium19). Our method outperforms state-of-the-art methods in most cases.

## 2 RELATED WORK

**Semi-supervised learning** (SSL) alleviates the issue of inadequacy of labelled data for training, which learns from both labelled and unlabelled data from predefined classes to get a strong classification model. Consistency-based approaches, including Mean-teacher Tarvainen & Valpola (2017), Mixmatch Berthelot et al. (2019) and Fixmatch Sohn et al. (2020), operate by enforcing model prediction consistency under various perturbations of the unlabelled data or over the course of training. Recent methods, such as Chen et al. (2020b;c; 2021), have shown improved SSL performance by introducing contrastive learning (*e.g.*, Chen et al. (2020a), He et al. (2020)). Several works Wang et al. (2022b); Rizve et al. (2022); Wang et al. (2024); Sun et al. (2024) extend the standard SSL to an open-world setting.

**Novel category discovery** (NCD) aims at categorizing unlabelled images from unseen classes by transferring knowledge from labelled data of seen classes Han et al. (2019). Various approaches have been proposed to address NCD, for example, Han et al. (2019) introduces a two-stage training method, which first utilizes metric learning, followed by learning to cluster the unlabelled data. Han et al. (2020; 2021); Zhao & Han (2021) utilize ranking statistics to generate pseudo positives among unlabelled novel classes. Zhong et al. (2021b) transfers semantic knowledge through MixUp augmentation between seen and novel classes, as well as reliable novel anchors with other examples. Zhong et al. (2021a) proposes a neighborhood contrastive loss and hard-negative generation process by mixing novel and seen classes. Fini et al. (2021) reformulates the NCD problem into classification based on dynamic class assignments using Sinkhorn-Knopp algorithm. Jia et al. (2021) addresses multi-modal NCD by inter- and intra-modal contrastive learning with permutation-ensembled ranking statistics. Gu et al. (2023) proposes a novel knowledge distillation framework, which utilizes our class-relation representation to regularize the learning of novel classes.

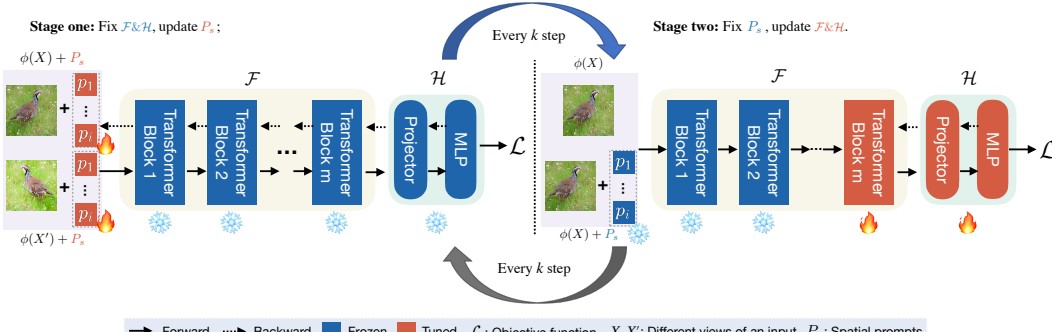

Figure 1: The overall framework of SPTNet. SPTNet alternates between data parameter tuning (*stage one*) and model parameter tuning (*stage two*). The data parameters are learnable prompts, for which we introduce spatial prompts $P_s$. The model parameters include the parameters of the top layer of the Transformer backbone $\mathcal{F}$ and a projection head $\mathcal{H}$.

**Generalized Category Discovery** (GCD) extends NCD by categorizing unlabelled images from both seen and unseen categories (Vaze et al. (2022)). Vaze et al. (2022) tackles this issue by tuning the representation of the pre-trained ViT model with DINO (Caron et al. (2021); Oquab et al. (2024)) with contrastive learning, followed by semi-supervised $k$-means clustering. ORCA Cao et al. (2022) considers the problem from a semi-supervised learning perspective and introduces an adaptive margin loss for better intra-class separability for both seen and unseen classes. CiPR Hao et al. (2024) introduces a method for more effective contrastive learning and a hierarchical clustering method for GCD without requiring the category number in the unlabelled data to be known a priori. SimGCD Wen et al. (2023) proposes a parametric method with entropy regularization to improve performance. DCCL Pu et al. (2023) improves clustering accuracy by alternating between estimating underlying visual conceptions and learning conceptional representations. They also introduce a dynamic conception generation and update mechanism to ensure consistent conception learning. PromptCAL Zhang et al. (2023) introduces a two-stage framework that iteratively generates and refines affinity graphs based on the model's current understanding of the data to enhance the semantic discriminativeness of pre-trained vision transformers. GPC Zhao et al. (2023) proposes a GMM-based method that can jointly learn robust representation for GCD and estimate the unknown category number. We also note the concurrent work Vaze et al. (2023) which improves GCD performance with a student-teacher mechanism.

**Prompt learning**, as the representative of data parameters learning methods, targets at simply prepending a few extra tokens to the input and provides an effective and efficient solution that matches the performance of fully fine-tuning, commonly used in Natural Language Processing (NLP). Recently, prompting learning has been used in vision tasks. Particularly, Visual Prompt Learning (VPT) Jia et al. (2022) has been introduced to optimize extra visual prompts on top of a pre-trained ViT backbone to achieve strong object recognition performance. Bahng et al. (2022) learns an additional "border" of input images as prompts to adapt large-scale pre-trained models, which improves the models' classification accuracy. There are also some works which utilize prompts to deal with different tasks, such as classification with imbalanced data Dong et al. (2022) or domain shift Wang et al. (2022a). Shtedritski et al. (2023); Khattak et al. (2023) offer the possibility of manipulating both textual and visual modalities through prompting.

## 3 METHODS

### 3.1 PRELIMINARIES

**Problem statement.** Assume that we have the open-world dataset $\mathcal{D}$, comprising two subsets: a labelled set $\mathcal{D}_l = \{(X_i, y_i)\}_{i=1}^{N_l} \subset \mathcal{X}_l \times \mathcal{Y}_l$ and an unlabelled set $\mathcal{D}_u = \{X_i\}_{i=1}^{N_u} \subset \mathcal{X}_u$, where $X_i \in \mathbb{R}^{3 \times H \times W}$. $H$ and $W$ are the height and width of the image. $\mathcal{Y}_l = \mathcal{C}_1$ and $\mathcal{Y}_u = \mathcal{C} = \mathcal{C}_1 \cup \mathcal{C}_2$ are the label space of labelled and the unlabelled samples. $\mathcal{C}, \mathcal{C}_1$, and $\mathcal{C}_2$ denote the label set for 'All', 'Old', and 'New' categories, respectively. The objective of GCD is to categorize all the unlabelled images in $\mathcal{D}_u$, having access to labels only in $\mathcal{D}_l$. For simplicity, hereafter we omit the subscript for each image $X_i$.

**Architecture.** We consider a parametric model consisting of a feature extractor $\mathcal{F}$ and a projection head $\mathcal{H}$. For an image $X$ from $\mathcal{D}$, we can obtain its class prediction by $\hat{y} = \mathcal{H}(\mathcal{F}(X))$. Specifically, we employ a Vision Transformer (ViT) Dosovitskiy et al. (2020) as the architecture. We consider

the Transformer encoder as $\mathcal{F}$ and a simple multilayer perceptron (MLP) as $\mathcal{H}$. In ViT, an image $X$ is first divided into $n$ patches $(x^1, \cdots, x^n)$, where $x^i \in \mathbb{R}^{3 \times h \times w}$ and $n = (H \times W)/(h \times w)$. The patches are then mapped into $d$-dimensional latent space by a shared linear projection layer e. Together with an extra learnable classification token *CLS*, the full model is formulated as:

$$
\begin{aligned}
(x^1, \cdots, x^n) &= \phi(X) \\
E_0 &= [\mathrm{e}(x^1); ...; \mathrm{e}(x^n)] \\
[CLS_i; E_i] &= L_i([CLS_{i-1}; E_{i-1}]) \\
\hat{y} &= \mathcal{H}(CLS_M),
\end{aligned}
\tag{1}
$$

where $\phi(\cdot)$ denotes the "pathcify" operator to divide the input image $X$ into patches $(x^1, \cdots, x^n)$, $L_i$ denotes the $i$-th layer of the ViT and $[\cdot]$ denotes concatenation.

**Model/Data parameters.** We consider optimizing both *model parameters* and *data parameters* for GCD. Optimizing the model parameters is the most common way to train or fine-tune a model by minimizing the loss function on a given dataset. Previous GCD methods, such as Vaze et al. (2022); Wen et al. (2023); Pu et al. (2023); Zhang et al. (2023), fine-tune the final transformer block and the linear projection layer of a pre-trained ViT model. In our method, the model parameters are in $\mathcal{F}$ and $\mathcal{H}$. Recently, visual prompt learning techniques have been introduced to effectively adapt pre-trained large-scale models to different downstream tasks, without the need of tuning the model parameters. We refer to such techniques as *optimizing data parameters*. Particularly, VPT Jia et al. (2022) inserts a sequence of learnable embeddings in the input for each Transformer encoder layer $L_i$ of the ViT model. Specifically, VPT freezes the feature extractor $\mathcal{F}$ but learns a set of prompt tokens $P_i = \{p_i^j; j = 1, \cdots, b\}$ with $p_i^j \in \mathbb{R}^d$ as part of the input for layer $L_i$. The input can be denoted as $[CLS_i; P_i; E_i]$. Instead of inserting several tunable parameters to each layer, Bahng et al. (2022) attaches learnable parameters $P_g$ to the border of the raw input image as $(x'^1, \cdots, x'^n) = \phi(X + P_g)$. We propose Spatial Prompt Tuning (SPT) for GCD, as will be described in Section 3.3, which attaches a learnable prompt for each image patch. Namely, we learn a set of prompts $P_s = \{p^j; j = 1, \cdots, n\}$ and attach the prompts to the input image patches by $(x^1 + p^1, \cdots, x^n + p^n) = \phi(X) + P_s$.

## 3.2 SPTNET: AN ALTERNATE PROMPT LEARNING FRAMEWORK FOR GCD

Large-scale pretraining (*e.g.*, DINO Caron et al. (2021) self-supervision) is the key ingredient in existing GCD methods Vaze et al. (2022); Wen et al. (2023); Pu et al. (2023); Zhang et al. (2023). As GCD requires learning from unlabelled data, contrastive self-supervised learning is the natural choice, which uses data augmentations to create different views of the same input image. These augmentations provide an inductive bias as to what is (not) semantically meaningful in an image. In this context, prompt tuning is a clear but unexplored option that enables efficient adaptation of pre-trained models. Our insight is that the learned prompt can also be used to generate a *novel view*, making it a suitable choice for the contrastive framework. Simultaneously optimizing the model and prompts seems appealing, but it results in instability and sub-optimal solutions for both data and model parameters.

To mitigate the issue, inspired by EM algorithm Dempster et al. (1977), we propose SPTNet, a two-stage alternative learning framework for GCD. The overall framework is illustrated in Fig. 1. The learning objective includes both representation learning and parametric classification, while our framework alternates between data parameter and model parameter optimization using the same learning objective.

Specifically, in vanilla contrastive learning, two different views, $X$ and $X'$, of the same input image are constructed as a positive pair. A set of other images are drawn from the dataset as negative samples $\mathcal{N}(X) = \{X_q^-; q = 1, \cdots, Q\}$. Then, the parameters of $\mathcal{F}$ can be updated by the InfoNCE loss Oord et al. (2018) using the data triplet $(X, X', \mathcal{N}(X))$:

$$
\mathcal{L}_{\mathrm{nce}}^{\mathrm{un}}(X, X', \mathcal{N}(X); \mathcal{F}, \tau_u) = -\log \frac{\exp(\cos(\mathcal{F}(X), \mathcal{F}(X'))/\tau_u)}{\sum_{q=1}^{Q} \exp(\cos(\mathcal{F}(X), \mathcal{F}(X_q^-))/\tau_u)},
\tag{2}
$$

where $\cos(\cdot, \cdot)$ is the cosine similarity between embedding feature vectors and $\tau_u$ is a temperature hyperparameter. Analogous to Eq. (2), supervised contrastive loss Khosla et al. (2020) $\mathcal{L}_{\mathrm{nce}}^{\mathrm{sup}}(X, \mathcal{P}(X), \mathcal{N}(X), y; \mathcal{F}, \tau_c)$ utilizes a set of positive samples $\mathcal{P}(X)$ having the same class label $y$ in the mini-batch.

Next, to assign labels to input instances, we use parametric methods to classify them into seen or new classes, as commonly done in image recognition. In supervised contrastive learning, this is

achieved through the simultaneous optimization of $\mathcal{F}$ and $\mathcal{H}$ using the cosine-softmax cross-entropy loss Gidaris & Komodakis (2018):

$$\mathcal{L}_{\text{cls}}^{\text{sup}}(X, y; \mathcal{H}, \mathcal{F}, \tau_s) = -\sum_{\kappa} y_\kappa \log \frac{\exp(\cos(\mathcal{H}(\mathcal{F}(X)), W_\kappa)/\tau_s)}{\sum_{\kappa'=1}^{|C|} \exp(\cos(\mathcal{H}(\mathcal{F}(X)), W_{\kappa'})/\tau_s)}, \tag{3}$$

where $\mathcal{H}(\mathcal{F}(X))$ and $W_\kappa$ are the $\ell_2$-normalized feature and the prototype vector of class $\kappa$ respectively. $X'$ is also used in the above loss, as an additional augmented version of $X$. For the unsupervised counterpart, $X'$ is removed from the input while the prediction $\hat{y} = \mathcal{H}(\mathcal{F}(X'))$ is used as a pseudo label for self-distillation. The loss can be denoted as $\mathcal{L}_{\text{cls}}^{\text{un}}(X, X'; \mathcal{H}, \mathcal{F}, \tau_t)$. Therefore, the overall loss $\mathcal{L}$ can be written as:

$$\mathcal{L} = (1 - \lambda)(\mathcal{L}_{\text{nce}}^{\text{un}} + \mathcal{L}_{\text{cls}}^{\text{un}}) + \lambda(\mathcal{L}_{\text{nce}}^{\text{sup}} + \mathcal{L}_{\text{cls}}^{\text{sup}}) - \epsilon\Delta, \tag{4}$$

where $\lambda, \epsilon$ are the balance factors, and $\Delta$ represents the the mean-entropy-maximisation regulariser Assran et al. (2022), computed by taking the entropy of the mean prediction of all samples in a mini-batch.

Our SPTNet alternates between optimizing data parameters and model parameters as follows:

***Stage one*: Fix $\mathcal{F}\&\mathcal{H}$ and update $P_s$.** In the first stage, we attach the same set of spatial prompts $P_s$ to the input images, $X$, $X'$, and $\mathcal{N}(X)$. The framework is trained with the loss in Eq. (4), while the image patches are replaced by their 'prompted' version. For example, $\phi(X)$ in Eq. (1) is replaced by $\phi(X) + P_s$, and the same applies to $X'$ and $\mathcal{N}(X)$. During training, we freeze the parameters of $\mathcal{F}\&\mathcal{H}$ and only update the prompt parameters of $P_s$. It is worth noting that, to facilitate the generalization, the weight decay for optimizing $P_s$ is set to zero to prevent prompts from being sparse. Meanwhile, during the learning in *Stage two*, our spatial prompting acts as a strong data augmentation. Increasing the variation in the parameters of $P_s$ leads to more diverse 'prompted' image pairs to benefit the representation learning. It was also noted in the literature that more diverse data augmentation is helpful for representation learning based on contrastive learning (*e.g.*, HaoChen et al. (2021)).

***Stage two*: Fix $P_s$ and update $\mathcal{F}\&\mathcal{H}$.** In the second stage, we freeze prompt parameters $P_s$ and learn the parameters of $\mathcal{H}$ and the top layer in $\mathcal{F}$, again, with the loss in Eq. (4). With our spatial prompt learning as a strong augmentation, we aim to obtain a representation that can better distinguish samples from different classes, as the core mechanism of contrastive learning involves implicitly clustering samples from the same class together.

Different from prior works that apply only hand-crafted augmentations, we propose to consider prompting the input with learnable prompts, *i.e.*, $\phi(X) + P_s$, as a new type of augmentation. The 'prompted' version of the input can be adopted by all loss terms. In this way, our framework can enjoy a learned augmentation that varies throughout the training process, enabling $\mathcal{F}$ to learn discriminative representations. Each stage optimizes the parameters for $k$ iterations.

### 3.3 SPATIAL PROMPT TUNING

Naively applying existing prompt tuning methods does not lead to satisfying performance (see $2^{nd}$ and $3^{rd}$ rows in Table 4). We speculate that prompts in the hidden model space rather than input space make it harder to align inputs within the contrastive framework, as evidenced by our empirical results in Fig. 5. Besides, a key insight in GCD is that object parts are effective in transferring knowledge between old and new categories Vaze et al. (2022).

Therefore, we propose Spatial Prompt Tuning (SPT) to serve as a learned data augmentation that enables the model to focus on local image object regions, while adapting the data representation from the pre-trained ViT model and maintaining the alignment with it. In SPT, we inject a small number of learnable parameters into the input image patches and keep the backbone $\mathcal{F}$ and the projection head $\mathcal{H}$ frozen during training. Unlike existing methods that introduce learnable tokens to hidden model space Jia et al.

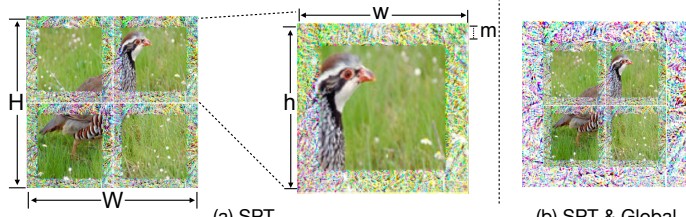

Figure 2: (a) An example of applying Spatial Prompt Tuning (SPT) to an image with a height $H$ and width $W$. For each image patch $x^j$ with a height $h$ and width $w$, we attach spatial prompts $P_s$ of size $m$ to it. (b) Joint spatial and global prompts for SPTNet.

(2022) or wrap prompts around the entire image border Bahng et al. (2022), SPT divides the image into patches and attaches learnable prompts to each partch. Specifically, let $(x^1, \cdots, x^n) = \phi(X)$ be the set of image patches divided from image $X$. For each patch $x^j \in \mathbb{R}^{3 \times h \times w}$, SPT wraps instance-agnostic prompts $P_s$ around it in a rectangular shape with a width of $m$, as illustrated in Fig. 2 (a). Thus, there are $6m(h + w - 2m)$ learnable parameters for the prompts of each patch.

Our SPTNet proceeds alternatively between the two stages and gradually learns the spatial prompts shared across all images. As revealed in Zhao & Han (2021), both global and local spatial information benefits novel category discovery. Therefore, apart from SPT tokens, SPTNet also wraps an additional global prompt around the entire image like Bahng et al. (2022), as illustrated in Fig. 2 (b).

## 4 EXPERIMENTS

### 4.1 EXPERIMENTAL SETUP

**Datasets.** We evaluate the effectiveness of SPT on three generic image recognition datasets (*i.e.*, CIFAR-10/100 Krizhevsky et al. (2009) and ImageNet-100 Tian et al. (2020)), three fine-grained datasets (*i.e.*, CUB Welinder et al. (2010), Stanford Cars Krause et al. (2013), and FGVC-Aircraft Maji et al. (2013)) contained in Semantic Shift Benchmark (SSB) Vaze et al. (2021), and the challenging large-scale fine-grained dataset Herbarium-19 Tan et al. (2019). For each dataset, we first subsample $|\mathcal{C}_1|$ seen (labelled) classes from all classes. Following Vaze et al. (2022), we subsample 80% samples in CIFAR-100 and 50% samples in all other datasets from the seen classes to construct $\mathcal{D}_l$, while the remaining images are treated as $\mathcal{D}_u$. The statistics of the datasets can be found in Table 1.

Table 1: Dataset statistics and training configurations.

| | Labelled | | Unlabelled | | Configs | | | | | |
|---|---|---|---|---|---|---|---|---|---|---|
| Dataset | #Num | #Class | #Num | #Class | $\mathrm{lr}_b$ | $\mathrm{wd}_b$ | $\mathrm{lr}_p$ | $\mathrm{wd}_p$ | $k$ | $m$ |
| CIFAR10 Krizhevsky et al. (2009) | 12.5K | 5 | 37.5K | 10 | 3e-3 | 5e-4 | 1.0 | 0 | 20 | 1 |
| CIFAR100 Krizhevsky et al. (2009) | 20.0K | 80 | 30.0K | 100 | 1e-3 | 5e-4 | 1.0 | 0 | 20 | 1 |
| ImageNet-100 Tian et al. (2020) | 31.9K | 50 | 95.3K | 100 | 3e-3 | 5e-4 | 10.0 | 0 | 20 | 1 |
| Herbarium 19 Tan et al. (2019) | 8.9K | 341 | 25.4K | 683 | 3e-3 | 5e-4 | 10.0 | 0 | 20 | 1 |
| CUB Welinder et al. (2010) | 1.5K | 100 | 4.5K | 200 | 0.05 | 5e-4 | 25.0 | 0 | 20 | 1 |
| Stanford Cars Krause et al. (2013) | 2.0K | 98 | 6.1K | 196 | 0.05 | 5e-4 | 25.0 | 0 | 20 | 1 |
| FGVC-Aircraft Maji et al. (2013) | 1.7K | 50 | 5.0K | 50 | 0.05 | 5e-4 | 25.0 | 0 | 20 | 1 |

**Evaluation protocol.** We use clustering accuracy ($ACC$) to evaluate the model performance, as per standard practice. During the evaluation, we compare the ground-truth labels $y_i$ with the predicted labels $\hat{y}_i$ and measure the ACC by $ACC = \frac{1}{|\mathcal{D}_u|} \sum_{i=1}^{|\mathcal{D}_u|} \mathbb{1}(y_i = \mathcal{G}(\hat{y}_i))$, where $\mathcal{G}$ represents the optimal permutation that gives the matching between the predicted labels with the ground truth.

**Implementation details.** We develop our SPTNet upon the SimGCD Wen et al. (2023) baseline and apply the spatial prompt tuning on the pre-trained ViT-B/16 backbone Caron et al. (2021). Specifically, we take the final feature corresponding to the *CLS* token from the backbone as the image feature, which has a dimension of 768. For the feature extractor $\mathcal{F}$, we only fine-tune the last block. We set the spatial prompt size $m$ to 1, while the globe prompt size to 30 which is the default value in Bahng et al. (2022). It is worth noting that our method yields extra parameters amounting to only 0.117% of those in the backbone architecture (see Appendix A for details). The two stages alternate every $k = 20$ iterations. All prompts are trained for 1,000 epochs with a batch size of 128. We utilize SGD as the optimizer for training, employing different learning rates ($\mathrm{lr}_p$, $\mathrm{lr}_b$) and weight decay parameters ($\mathrm{wd}_p$, $\mathrm{wd}_b$) to update prompts and the model. The training hyper-parameters, determined on the validation data splits, are shown in Table 1. We set the balancing factor $\lambda$ to 0.35 and the temperature values $\tau_u$ and $\tau_c$ to 0.07 and 1.0, respectively, following Wen et al. (2023). For the temperature values $\tau_t$ and $\tau_s$ in the classification losses, we also set them to 0.07 and 0.1. All experiments are conducted using an NVIDIA GeForce RTX 3090 GPU.

### 4.2 MAIN RESULTS

**Evaluation on generic datasets.** We evaluate SPTNet on three generic datasets, CIFAR-10, CIFAR-100 and ImageNet-100. We compare SPTNet with previous state-of-the-art methods and two concurrent methods (DCCL Pu et al. (2023) and PromptCAL Zhang et al. (2023)). The results are shown in Table 2. We can see that our method consistently outperforms previous state-of-the-art methods. Specifically, SPTNet surpasses the baseline SimGCD by 0.4% on CIFAR-10, 1.9% on CIFAR-100, and 2.5% on ImageNet-100 for 'All' classes; it also outperforms concurrent methods on both CIFAR-100 and ImageNet-100. SPTNet performs on par with PromptCAL on CIFAR-10 but with much fewer learnable parameters and shorter training time (see Table 13). Note that for CIFAR-10 and

CIFAR-100, the images are of extremely low-resolution ($32 \times 32$). As such, limited information is provided in each patch, leading to limited gains from our proposed (local) spatial prompting. On ImageNet-100, performance boosts are difficult to yield, as the original DINO backbone is already highly tuned for this dataset. This is evidenced by the gains (usually) being substantially less between the previous state-of-the-art and the simple $k$-means on raw DINO features Vaze et al. (2022).

Table 2: Evaluation on three generic image recognition datasets. Bold values represent the best results, while underlined values represent the second best results.

| Method | CIFAR-10 | | | CIFAR-100 | | | ImageNet-100 | | |
|---|---|---|---|---|---|---|---|---|---|
| | All | Old | New | All | Old | New | All | Old | New |
| $k$-means Arthur & Vassilvitskii (2006) | 83.6 | 85.7 | 82.5 | 52.0 | 52.2 | 50.8 | 72.7 | 75.5 | 71.3 |
| RankStats+ Han et al. (2021) | 46.8 | 19.2 | 60.5 | 58.2 | 77.6 | 19.3 | 37.1 | 61.6 | 24.8 |
| UNO+ Fini et al. (2021) | 68.6 | **98.3** | 53.8 | 69.5 | 80.6 | 47.2 | 70.3 | **95.0** | 57.9 |
| GCD Vaze et al. (2022) | 91.5 | 97.9 | 88.2 | 73.0 | 76.2 | 66.5 | 74.1 | 89.8 | 66.3 |
| ORCA Cao et al. (2022) | 96.9 | 95.1 | 97.8 | 74.2 | 82.1 | 67.2 | 79.2 | 93.2 | 72.1 |
| SimGCD Wen et al. (2023) | 97.1 | 95.1 | 98.1 | 80.1 | 81.2 | **77.8** | 83.0 | 93.1 | 77.9 |
| DCCL Pu et al. (2023) | 96.3 | 96.5 | 96.9 | 75.3 | 76.8 | 70.2 | 80.5 | 90.5 | 76.2 |
| PromptCAL Zhang et al. (2023) | **97.9** | 96.6 | 98.5 | 81.2 | 84.2 | 75.3 | 83.1 | 92.7 | 78.3 |
| **SPTNet (Ours)** | 97.3 | 95.0 | **98.6** | **81.3** | **84.3** | 75.6 | **85.4** | 93.2 | **81.4** |

**Evaluation on fine-grained datasets.** Table 3 presents the results on fine-grained datasets including the SSB benchmark and Herbarium 19 dataset. The unsatisfactory performance of $k$-means and ORCA highlights the difficulty in discovering fine-grained categories due to large intra-class and small inter-class variations. In contrast, SPTNet demonstrates superior performance to SimGCD, DCCL, and PromptCAL, achieving an average absolute improvement of ∼5% and an average proportional improvement of ∼10% across all evaluated datasets in SSB, specifically on 'All' classes. As there is a clear semantic axis in SSB benchmark, and data augmentations implicitly define this 'semantic axis' or taxonomy in contrastive learning, SPT as a learned data augmentation ultimately enhances the GCD performance. This indicates that global and local prompts assist the model in focusing on details that dominate correctness in fine-grained recognition in GCD.

Table 3: Evaluation on the Semantic Shift Benchmark (SSB) and Herbarium 19. Bold values represent the best results, while underlined values represent the second best results.

| Method | CUB | | | Stanford Cars | | | FGVC-Aircraft | | | Herbarium19 | | |
|---|---|---|---|---|---|---|---|---|---|---|---|---|
| | All | Old | New | All | Old | New | All | Old | New | All | Old | New |
| $k$-means Arthur & Vassilvitskii (2006) | 34.3 | 38.9 | 32.1 | 12.8 | 10.6 | 13.8 | 12.9 | 12.9 | 12.8 | 13.0 | 12.2 | 13.4 |
| RankStats+ Han et al. (2021) | 33.3 | 51.6 | 24.2 | 28.3 | 61.8 | 12.1 | 27.9 | 55.8 | 12.8 | 27.9 | 55.8 | 12.8 |
| UNO+ Fini et al. (2021) | 35.1 | 49.0 | 28.1 | 35.5 | 70.5 | 18.6 | 28.3 | 53.7 | 14.7 | 28.3 | 53.7 | 14.7 |
| GCD Vaze et al. (2022) | 51.3 | 56.6 | 48.7 | 39.0 | 57.6 | 29.9 | 45.0 | 41.1 | 46.9 | 35.4 | 51.0 | 27.0 |
| ORCA Cao et al. (2022) | 36.3 | 43.8 | 32.6 | 31.9 | 42.2 | 26.9 | 31.6 | 32.0 | 31.4 | 20.9 | 30.9 | 15.5 |
| SimGCD Wen et al. (2023) | 60.3 | 65.6 | 57.7 | 53.8 | 71.9 | 45.0 | 54.2 | 59.1 | 51.8 | 43.0 | 58.0 | 35.1 |
| DCCL Pu et al. (2023) | 63.5 | 60.8 | 64.9 | 43.1 | 55.7 | 36.2 | - | - | - | - | - | - |
| PromptCAL Zhang et al. (2023) | 62.9 | 64.4 | 62.1 | 50.2 | 70.1 | 40.6 | 52.2 | 52.2 | 52.3 | 37.0 | 52.0 | 28.9 |
| **SPTNet (Ours)** | **65.8** | 68.8 | **65.1** | **59.0** | **79.2** | 49.3 | **59.3** | **61.8** | **58.1** | **43.4** | **58.7** | **35.2** |

## 4.3 ABLATION STUDY

In this part, we primarily focus on the challenging SSB to assess the effectiveness of different components and report the averaged results among CUB, Stanford Cars, and FGVC-Aircraft. As our method employs the same pre-trained model and objective function as SimGCD, we consider SimGCD as the baseline method for comparison.

**Effect of prompt-related techniques.** We first experiment with the strong SimGCD baseline using the recommended configuration in Wen et al. (2023). Table 4 presents the results of the ablation study on the components of our SPT. The $2^{nd}$ row shows the performance of adopting the VPT method on the pre-trained SimGCD. Comparing with the raw SimGCD base in the $1^{st}$ row, the performance after adopting VPT is dropped. After adopting the global prompt ($3^{rd}$ row) on the pre-trained SimGCD, the performance is increased by 0.6% on 'All' classes. This indicates that naively applying existing prompt tuning methods does not yield satisfactory performance on GCD; the improvement by the global prompt, though marginal, is still encouraging, as it suggests that the pixel-level prompt method is suitable for the GCD setting when compared with VPT. Adopting our SPT ($4^{th}$ row) on pre-trained SimGCD gives a relatively larger improvement of 1.8% on 'All' classes. The effectiveness of our proposed method may be attributed to the spatial design for exploring semantic discrimination in local regions. Our alternate training strategy ($5^{th}$ & $7^{th}$ rows) can effectively improve the performance, demonstrating its effectiveness. We also explore a variant of SPT by using *shared* prompts across all patches ($6^{th}$ row), which also demonstrates promising performance. After further introducing the global prompts ($7^{th}$ & $8^{th}$ row), the performance is further improved. The $8^{th}$ row corresponds to our default SPTNet, which achieves the best performance. We refer to the variant in the $5^{th}$ row as SPTNet-S (Shared), and the variant in the $6^{th}$ row as SPTNet-P (Patch). Both of these variants are

Table 4: Comparison on effectiveness of different prompting methods on SSB. We report the average test accuracy score over all component datasets of SSB (*i.e.*, CUB, Stanford Cars and FGVC-Aircraft). 'Shared' and 'Alter' refer to a single *shared* prompt for all patches and *alternative* learning. Row (9) represents SPTNet and rows (5) and (6) represent its two variants SPTNet-S and SPTNet-P.

| No | Method config | Prompt config | All | Old | New |
|---|---|---|---|---|---|
| (1) | | None (baseline) | 56.1 | 65.5 | 51.5 |
| (2) | SimGCD Wen et al. (2023) | +VPT Jia et al. (2022) | $54.4^{-1.7}$ | $64.7^{-0.8}$ | $49.1^{-2.4}$ |
| (3) | | +Global Bahng et al. (2022) | $56.7^{+0.6}$ | $64.6^{-0.9}$ | $53.5^{+2.0}$ |
| (4) | | +SPT | $57.9^{+1.8}$ | $67.2^{+1.7}$ | $53.3^{+1.8}$ |
| (4) | | +Global Bahng et al. (2022) | $57.8^{+1.7}$ | $66.3^{+0.8}$ | $53.8^{+2.3}$ |
| (5) | +Alter | +Shared | $60.5^{+4.4}$ | $68.6^{+3.1}$ | $56.5^{+5.0}$ |
| (6) | | +SPT | $59.1^{+3.0}$ | $68.5^{+3.0}$ | $54.5^{+3.0}$ |
| (7) | +Alter | +Shared & Global Bahng et al. (2022) | $60.9^{+4.8}$ | $69.0^{+3.5}$ | $57.3^{+5.8}$ |
| (8) | | +SPT & Global Bahng et al. (2022) | $61.4^{+5.3}$ | $69.9^{+4.4}$ | $57.5^{+6.0}$ |

relatively more parameter efficient (see Appendix A for details), spatially SPTNet-S, while obtaining superior performance (more results of these two variants can be found in Appendix C).

**Effect of different training strategies.** To investigate the impact of different training strategies, we conduct additional experiments on both generic and fine-grained datasets. We consider two different training strategies, namely, (i) *end-to-end* ($3^{rd}$ row): both the data parameters and the model parameters are jointly trained in an end-to-end fashion; (ii) *data first* ($4^{th}$ row): the prompt parameters are optimized first, followed by the model parameters; (iii) *model first* ($5^{th}$ row): the model parameters are optimized first, followed by the prompt parameters; and (iv) *alternative* ($6^{th}$ row): our alternative training strategy, which optimizes model and data parameters alternatively, every other $k$ iterations. The results are presented in Table 5. Comparing rows (3)-(6) with the SimGCD baseline in row (1), we can see that SPTNet consistently outperforms SimGCD and our alternative training strategy leads to the best performance. Since the SPTNet is built upon the pre-trained SimGCD, one might wonder about the performance of further fine-tuning SimGCD. In the $2^{nd}$ row, we show the results after further fine-tuning the pre-trained SimGCD. An improvement can be achieved, while the margin is significantly smaller compared with the improvement achieved by SPTNet. This suggests that both our SPT and alternate training strategy are beneficial for GCD.

Table 5: Evaluation on ImageNet-100 and SSB using different training strategies.

| No | Methods | ImageNet-100 | | | SSB | | |
|---|---|---|---|---|---|---|---|
| | | All | Old | New | All | Old | New |
| (1) | SimGCD Wen et al. (2023) | 83.0 | 93.1 | 77.9 | 56.1 | 65.5 | 51.5 |
| (2) | SimGCD (further fine-tune) | 84.3 | 93.1 | 79.7 | 57.0 | 66.0 | 52.3 |
| (3) | SPTNet (end-to-end) | 84.1 | 92.8 | 80.0 | 58.6 | 67.4 | 53.2 |
| (4) | SPTNet (data first) | 83.5 | 92.9 | 77.7 | 58.0 | 66.4 | 51.9 |
| (5) | SPTNet (model first) | 84.8 | 93.3 | 80.6 | 59.2 | 67.8 | 54.9 |
| (6) | SPTNet (alternative) | **85.4** | **93.2** | **81.4** | **61.4** | **69.9** | **57.5** |

**Effects of alternating frequency and prompt size.** In our alternative training strategy, we alternate between the data and model parameter optimization every other $k$ iterations. Meanwhile, we also need to determine the spatial prompt size $m$. In Fig. 3, we present the average ACC on SSB with varying $k$ and $m$ respectively. For $k$, we do not observe significant differences among different choices, and thus use a moderate value of 20 as our default choice. For $m$, we find that a smaller value generally leads to better performance. When $m$ is too large, the image content might be over-occluded, causing difficulty for the model to properly recognize the object. We also show the effects of global prompt size in Appendix D.

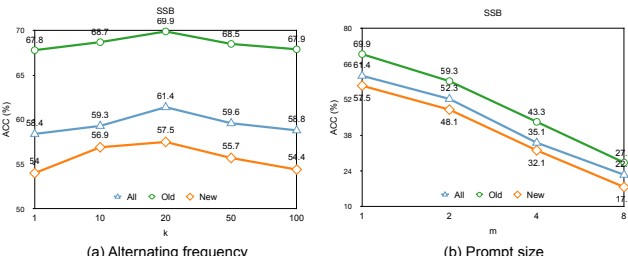

(a) Alternating frequency          (b) Prompt size

Figure 3: Effects of different choices of alternating frequency (a) and prompt size (b) on SSB (*i.e.*, CUB, Stanford Cars and FGVC-Aircraft). We report the averaged results and show the influence on 'All', 'Old' and 'New' classes.

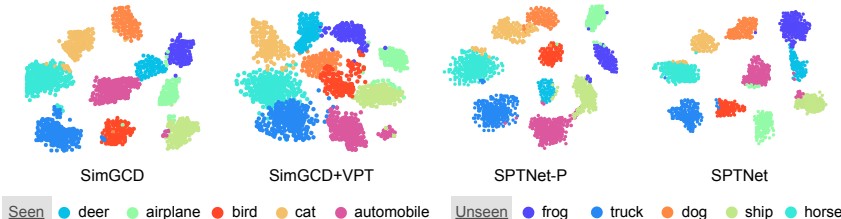

Figure 4: t-SNE visualization of representations on CIFAR-10. SPTNet produces the most discriminative representations among all compared methods.

### 4.4 QUALITATIVE COMPARISON

**How do prompts affect the representations?** To investigate the influence of different visual prompts, we visualize representations on CIFAR-10 through t-SNE Van der Maaten & Hinton (2008) in Fig. 4. We compare representations of the SimGCD baseline, SimGCD+VPT, SPTNet, as well as SPTNet-P (which contains only the spatial prompts). They correspond to the models in row (1), row (2), row (8), and row (6) in Table 4. Comparing the representations of SimGCD and SimGCD+VPT, VPT appears to have a negative impact on the representation, leading to clutter between seen and unseen classes (*e.g.*, bird and dog) in the GCD setting. This is also aligned with the deteriorated performance of SimGCD+VPT in Table 4. Both SPTNet-P and SPTNet produce more discriminative features and more compact clusters than SimGCD. Thanks to the global prompt, SPTNet further improves the representation over STPNet-P.

**How do prompts affect the model's attention?** The attention map provides very helpful clues to understanding the Transformer-based models' focus on the input. We extract the attention maps for the *CLS* token from different attention heads in the last layer of the ViT backbone and show the top 10% most attended patches in Fig. 5. We observe that for SimGCD and SimGCD+Global (*i.e.*, row (4) in Table 4), different heads may focus on the same region (*e.g.*, in the 'seen' example, $h_2/h_3/h_{10}$ of SimGCD and $h_4/h_5/h_9$ of SimGCD+Global) and some heads may attend to the background regions (*e.g.*, in the 'unseen' example, $h_2/h_4/h_7$ for SimGCD and $h_1/h_5$ for SimGCD+Global). In contrast, SPT and SPT&Global attend to more diverse regions of the object and focus more on the foreground object regions, with the latter performing better.

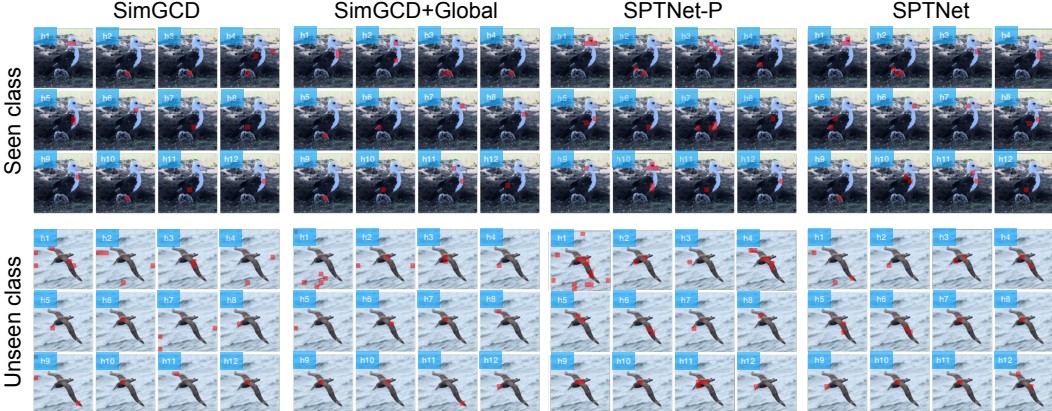

Figure 5: Attention visualization of different heads (numbered as $h_1$ to $h_{12}$). The top 10% attended patches are shown in red.

More results and analysis can be found in the Appendix.

## 5 CONCLUSION

In conclusion, we have introduced SPTNet, an efficient framework for Generalized Category Discovery (GCD). We propose a two-stage alternative optimization scheme, optimizing both model and data parameters, to enhance alignment between the pre-trained model and the target task. Additionally, we introduce spatial prompt tuning (SPT) as a method to focus on object parts and facilitate knowledge transfer between seen and unseen classes. Experimental evaluations demonstrate the superiority of SPTNet over existing methods.

**Acknowledgements** This work is supported by Hong Kong Research Grant Council - Early Career Scheme (Grant No. 27208022), National Natural Science Foundation of China (Grant No. 62306251), and HKU Seed Fund for Basic Research.

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

## APPENDIX

## A    DISCUSSION ON DIFFERENT VARIANTS OF SPTNET

As discussed in Section 3.3, our default SPTNet has both spatial and global prompts. In Section 4.3, we also introduce two variants of SPTNet with reduced prompt parameters, SPTNet-P (Patch) and SPTNet-S (Shared). SPTNet-P attaches only the spatial prompts without the global prompt to the input (row 6 in Table 4). The spatial prompts vary for different patches. SPTNet-S attaches a single shared spatial prompt without the global prompt to the input (row 7 in Table 4). In Figure 6, we compare the prompts of SPTNet, SPTNet-P and SPTNet-S.

As SPT wraps a small number of parameters around the raw input image in a rectangular shape with a width of $m$. As also discussed in Section 3.3, the number of parameters for the spatial prompt of each patch is $6m(h + w - 2m)$. Let $h = w = 16$, $m = 1$, and the number of patches $n = 196$. The number of parameters for a single spatial prompt is $6 \times 1 \times (16 + 16 - 2) = 5,880$. 196 such prompts give $196 \times 5,880 = 35,280 \approx 0.034$M parameters. As for the global padding, the number of parameters is $6m^+(H + W - 2m^+)$. Let $H = W = 224$. The number of parameters for the global prompt is $6 \times 30 \times (224 + 224 - 60) = 69,840$. Therefore, the total number of parameters for SPT & Global is $35,280 + 69,840 = 105,120$. As the backbone model, ViT-B, has 86M parameters, SPTNet, SPTNet-P, and SPTNet-S only introduce 0.117%, 0.039%, and 0.0002% extra parameters compared to ViT-B, respectively.

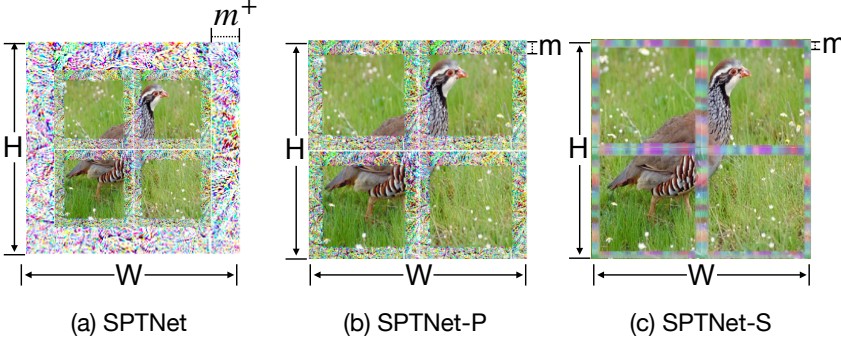

|      (a) SPTNet      |      (b) SPTNet-P      |      (c) SPTNet-S      |

Figure 6: Prompts of SPTNet, SPTNet-P and SPTNet-S. SPTNet has both distinct spatial prompts and a global prompt; SPTNet-P has multiple distinct spatial prompts; SPTNet-S has a single shared spatial prompt.

## B  TRAINING CONFIGURATIONS FOR SPTNET-P AND SPTNET-S

We show the training configurations for SPTNet-P and SPTNet-S in Table 6 and Table 7 respectively. The prompt size of SPT $m$ is set to 1 by default for both SPTNet-P and SPTNet-S. We set the global prompt size $m^+$ in SPTNet to 30 (see Appendix D). For the ViT model, specifically, we resize all the input images into $224 \times 224$, so we have $h \times w = 14 \times 14 = 196$ patches with a resolution of $16 \times 16$ pixels.

Table 6: Training configurations for SPTNet-P / SPTNet-S.

| Dataset | Configs | | | | | |
|---|---|---|---|---|---|---|
| | $\mathtt{lr}_b$ | $\mathtt{wd}_b$ | $\mathtt{lr}_p$ | $\mathtt{wd}_p$ | $k$ | $m$ |
| CIFAR10 Krizhevsky et al. (2009) | 3e-3 | 5e-4 | 20.0 | 0 | 20 | 1 |
| CIFAR100 Krizhevsky et al. (2009) | 1e-3 | 5e-4 | 1.0 | 0 | 20 | 1 |
| ImageNet-100 Tian et al. (2020) | 3e-3 | 5e-4 | 10.0 | 0 | 20 | 1 |
| Herbarium 19 Tan et al. (2019) | 5e-3 | 5e-4 | 1.0 | 0 | 20 | 1 |
| CUB Welinder et al. (2010) | 0.05 | 5e-4 | 25.0 | 0 | 20 | 1 |
| Stanford Cars Krause et al. (2013) | 0.05 | 5e-4 | 10.0 | 0 | 20 | 1 |
| FGVC-Aircraft Maji et al. (2013) | 0.05 | 5e-4 | 1.0 | 0 | 20 | 1 |

Table 7: Training configurations for SPTNet.

| Dataset | Configs | | | | | | |
|---|---|---|---|---|---|---|---|
| | $\mathtt{lr}_b$ | $\mathtt{wd}_b$ | $\mathtt{lr}_p$ | $\mathtt{wd}_p$ | $k$ | $m$ | $m^+$ |
| CIFAR10 Krizhevsky et al. (2009) | 3e-3 | 5e-4 | 1.0 | 0 | 20 | 1 | 30 |
| CIFAR100 Krizhevsky et al. (2009) | 3e-3 | 5e-4 | 5.0 | 0 | 20 | 1 | 30 |
| ImageNet-100 Tian et al. (2020) | 3e-3 | 5e-4 | 5.0 | 0 | 20 | 1 | 30 |
| Herbarium 19 Tan et al. (2019) | 5e-3 | 5e-4 | 1.0 | 0 | 20 | 1 | 30 |
| CUB Welinder et al. (2010) | 0.05 | 5e-4 | 25.0 | 0 | 20 | 1 | 30 |
| Stanford Cars Krause et al. (2013) | 0.05 | 5e-4 | 10.0 | 0 | 20 | 1 | 30 |
| FGVC-Aircraft Maji et al. (2013) | 0.05 | 5e-4 | 1.0 | 0 | 20 | 1 | 30 |

## C    BENCHMARKING RESULTS OF SPTNET-P AND SPTNET-S

We further evaluate the performance of the more parameter-efficient SPTNet variants, SPTNet-P and SPTNet-S, in Table 8 and Table 9. As can be seen, SPTNet, SPTNet-P and SPTNet-S consistently outperform the baseline in all cases.

Table 8: Evaluation on three generic image recognition datasets. Bold values represent the best results, while underlined values represent the second best results.

| Method | CIFAR-10 | | | CIFAR-100 | | | ImageNet-100 | | |
|---|---|---|---|---|---|---|---|---|---|
| | All | Old | New | All | Old | New | All | Old | New |
| SimGCD Wen et al. (2023) | 97.1 | 95.1 | 98.1 | 80.1 | 81.2 | **77.8** | 83.0 | 93.1 | 77.9 |
| **SPTNet-P (Ours)** | **97.5** | 95.2 | 98.5 | **82.0** | **85.5** | 75.0 | **85.5** | 93.9 | 81.2 |
| **SPTNet-S (Ours)** | **97.5** | **95.9** | 98.3 | 81.0 | 83.8 | 75.4 | **85.5** | **94.1** | 81.2 |
| **SPTNet (Ours)** | 97.3 | 95.0 | **98.6** | 81.3 | 84.3 | 75.6 | 85.4 | 93.2 | **81.4** |

Table 9: Evaluation on the Semantic Shift Benchmark (SSB) and Herbarium 19. Bold values represent the best results, while underlined values represent the second best results.

| Method | CUB | | | Stanford Cars | | | FGVC-Aircraft | | | Herbarium19 | | |
|---|---|---|---|---|---|---|---|---|---|---|---|---|
| | All | Old | New | All | Old | New | All | Old | New | All | Old | New |
| SimGCD Wen et al. (2023) | 60.3 | 65.6 | 57.7 | 53.8 | 71.9 | 45.0 | 54.2 | 59.1 | 51.8 | 43.0 | 58.0 | 35.1 |
| **SPTNet-P (Ours)** | 64.6 | **70.5** | 61.6 | 55.6 | 74.4 | 46.5 | 57.2 | 60.6 | 55.5 | 43.3 | 58.0 | **35.5** |
| **SPTNet-S (Ours)** | 65.0 | 69.1 | 62.9 | **60.1** | 75.3 | **52.8** | 56.3 | 61.4 | 53.8 | **43.4** | 58.6 | 35.2 |
| **SPTNet (Ours)** | **65.8** | 68.8 | **65.1** | 59.0 | **79.2** | 49.3 | **59.3** | **61.8** | **58.1** | **43.4** | **58.7** | 35.2 |

# D  EFFECTS OF GLOBAL PROMPT SIZE $m^+$

As the global prompt size $m^+$ may affect the performance of SPTNet, we experiment with different global prompt sizes, namely $m^+ = 1, 10, 20, 30, 40, 50$. We measure accuracy on the CUB dataset using the same architecture and configurations in the main paper. Fig. 7 demonstrates that $m^+ = 30$ yields the best overall performance, which is the default setting in our main paper.

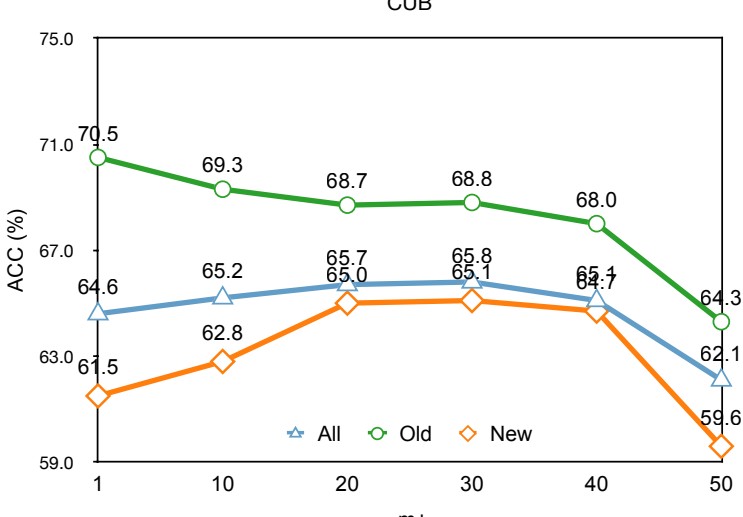

Figure 7: Performance of SPTNet with different global prompt sizes $m^+$ on CUB. We show the influence on 'All', 'Old' and 'New' classes. When $m^+$ is set to 30, SPTNet achieves the best performance on 'All' categories.

# E    VISUALIZATION OF LEARNED PROMPTS

We visualize different prompts after convergence in Fig. 8. Except for the end-to-end strategy, SPT and its variants commonly exhibit active parameters in prompts. This is attributed to their instance-agnostic nature, which enables them to handle variations in object locations across the dataset. Consequently, parameter values do not degrade to zero for a region patch that is background in one input but foreground in another. It is also worth noting that most prompts (particularly located at the borders) are deactivated when employing the end-to-end strategy. We hypothesize that this is due to the network unintentionally adopting a "shortcut" approach, where it only updates model parameters to achieve invariance to learned augmentation while optimizing both model and data (prompt) parameters simultaneously. This also validates the need for alternate training when learned augmentation is applied.

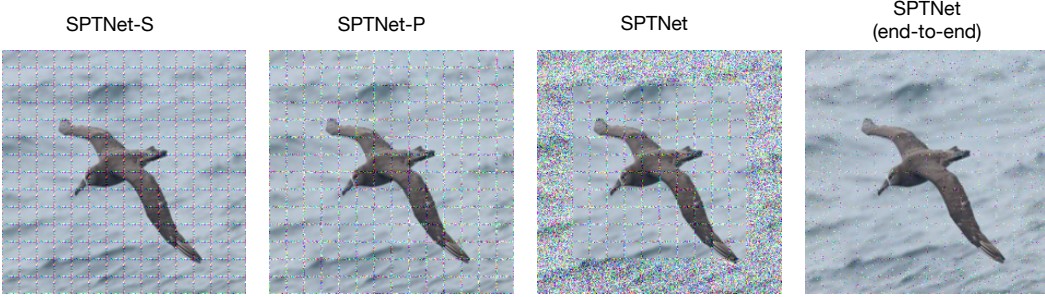

Figure 8: Visualization of different learned prompts. Parameters of SPT and its variants are mostly active, while most prompts are much less active when employing the end-to-end strategy.

# F   RESULTS BASED ON DINOv2

Here, we replace the pre-trained DINO model with the recently improved DINOv2 model Oquab et al. (2024) which is empowered with stronger representation capacity from unlabelled data. Results are shown in Table 10. We can observe that the stronger DINOv2 representation indeed enhances model performance as expected, especially in the 'New' categories, and SPTNet and the two variants still consistently outperform SimGCD.

Table 10: Evaluation on CUB and ImageNet-100 using the pre-trained DINOv2 model. Bold values represent the best results, while underlined values represent the second best results.

|  | CUB | | | ImageNet-100 | | |
| --- | --- | --- | --- | --- | --- | --- |
| Method | All | Old | New | All | Old | New |
| DINO+SimGCD Wen et al. (2023) | 60.3 | 65.6 | 57.7 | 83.0 | 93.1 | 77.9 |
| DINO+**SPTNet-P (Ours)** | 64.6 | **70.5** | 61.6 | 85.5 | 93.9 | 81.2 |
| DINO+**SPTNet-S (Ours)** | 65.0 | 69.1 | 62.9 | 85.5 | 94.1 | 81.2 |
| DINO+**SPTNet (Ours)** | 65.8 | 68.8 | 65.1 | 85.4 | 93.2 | 81.4 |
| DINOv2+SimGCD Wen et al. (2023) | 67.4 | 69.5 | 66.3 | 88.5 | 96.2 | 84.6 |
| DINOv2+**SPTNet-P (Ours)** | 69.0 | 69.7 | 68.7 | 90.5 | 96.3 | 87.5 |
| DINOv2+**SPTNet-S (Ours)** | 69.0 | 70.3 | 68.3 | **90.6** | **96.4** | **87.6** |
| DINOv2+**SPTNet (Ours)** | **69.2** | 69.0 | **69.3** | 90.1 | 96.1 | 87.1 |

## G    ROBUSTNESS OF SPTNET FOR GCD WITH DOMAIN SHIFTS

To validate the robustness of SPTNet, we test our method in a more challenging setting, GCD with domain shifts. We conduct experiments on the largest UDA dataset DomainNet Peng et al. (2019), containing about 0.6 million images with 345 categories distributed among six domains. We apply the data construction process in Vaze et al. (2022) to construct the'Old', 'New' and 'All' splits based on DomainNet and evaluate different methods on our constructed data. To account for domain shifts that were not considered in Vaze et al. (2022), we construct the partially labelled data by using images from both the 'real' domain and the 'painting' domain to train the model. Specifically, we utilise a subset of labelled images from select classes in the 'real' domain, along with unlabelled images from all classes in the 'painting' domain. We assess the model's performance on both the 'real' and 'painting' domains. Additionally, we evaluate the model on images from other previously unseen domains, including 'quickdraw', 'sketch', 'infograph', and 'clipart'. Results are shown in Table 11. Compared with other baseline methods in the vanilla GCD setting, we find that SPTNet can perform well on (i) labelled and seen domain (*i.e.*, 'real'), (ii) unlabelled but seen domains (*i.e.*, 'painting') and (iii) unseen domains (*i.e.*, others, including 'quickdraw', 'sketch', 'infograph', and 'clipart').

Table 11: Evaluation on the DomainNet benchmark. The model is trained on the 'real' and 'painting' domains and we report the respective results on real, painting and the remaining four domains (*i.e.*, others). Bold values represent the best results, while underlined values represent the second best results.

| Methods | Real | | | Painting | | | Others | | |
|---|---|---|---|---|---|---|---|---|---|
| | All | Old | New | All | Old | New | All | Old | New |
| RankStats+ | 34.1 | 61.9 | 19.7 | 29.7 | **49.7** | 9.6 | 14.3 | **25.5** | 5.5 |
| UNO+ | 44.2 | 72.2 | 29.7 | 30.1 | 45.1 | 17.2 | 14.0 | 23.4 | 7.4 |
| ORCA | 31.9 | 49.8 | 23.5 | 28.7 | 38.5 | 7.1 | 10.4 | 19.5 | 8.1 |
| GCD | 47.3 | 53.6 | 44.1 | 32.9 | 41.8 | 23.0 | 15.2 | 22.0 | 11.1 |
| SimGCD | 61.3 | **77.8** | 52.9 | 34.5 | 35.6 | 33.5 | 16.7 | 22.5 | 12.2 |
| **SPTNet (Ours)** | **63.1** | 75.9 | **56.4** | **39.2** | 43.1 | **35.2** | **17.4** | 22.2 | **13.6** |

# H    UNKNOWN CATEGORY NUMBER

As the total number of categories (GT) cannot be accessed in the real-world setting, we evaluate our SPTNet-P with an estimated number of categories using an off-the-shelf method Vaze et al. (2022) (see Table 12) We also evaluate our method with varying category numbers (see Fig. 9). Our evaluation includes two representative datasets: CUB for fine-grained and ImageNet-100 for generic classification tasks. We find that our method consistently outperforms SimGCD on both datasets when the exact number of categories is unknown.

Table 12: Performance of SPTNet-P and the baseline method SimGCD with an estimated number of categories on CUB and ImageNet-100. Bold values represent the best results.

| Method | $|\mathcal{C}|$ | CUB | | | ImageNet-100 | | |
|---|---|---|---|---|---|---|---|
| | | All | Old | New | All | Old | New |
| SimGCD Wen et al. (2023) | GT (200/100) | 60.3 | 65.6 | 57.7 | 83.0 | 93.1 | 77.9 |
| **SPTNet-P (Ours)** | GT (200/100) | 64.6 | 70.5 | 61.6 | **85.5** | **93.9** | **81.2** |
| SimGCD Wen et al. (2023) | Est. (231/109) | 61.0 | 66.0 | 58.6 | 81.1 | 90.9 | 76.1 |
| **SPTNet-P (Ours)** | Est. (231/109) | **65.2** | **71.0** | **62.3** | 83.4 | 91.8 | 74.6 |

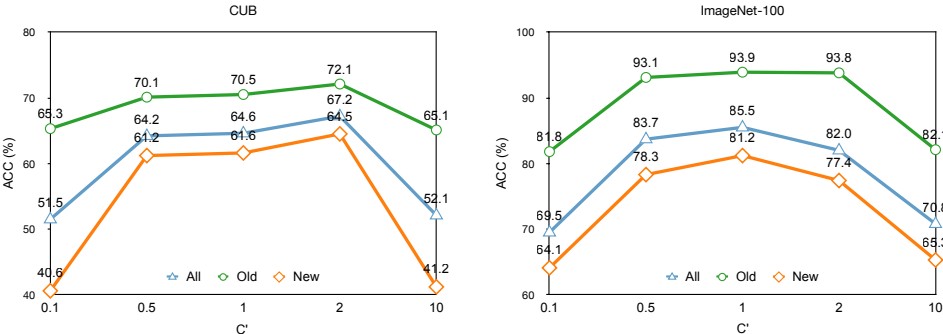

Figure 9: Performance with varying category numbers. We experiment with category numbers obtained by multiplying the GT number with different factors $C' = \{0.1, 0.5, 1.0, 2.0, 10.0\}$.

We also evaluate our SPTNet-P on CIFAR-100 dataset with fewer known categories. The results are shown in Fig. 10, where 50% of the samples from known classes are labelled. The results indicate that SPTNet is robust in few-class scenarios and outperforms the concurrent method, PromptCal Zhang et al. (2023). It is more challenging for models to infer novel semantic clustering when fewer classes are known due to semantic shifts, resulting in decreased performance for all methods. This further demonstrates the effectiveness of our proposed method.

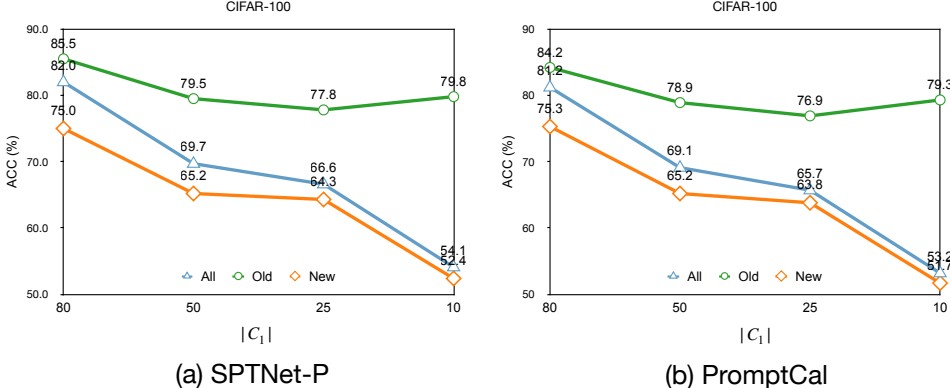

(a) SPTNet-P                    (b) PromptCal

Figure 10: Performance with a varying number of known classes $|\mathcal{C}_1|$.

# I PERFORMANCE AND TIME EFFICIENCY

To assess the practicality of different methods, we conducted further comparisons in terms of accuracy, training time per epoch, and inference time. The results are presented in Table 13. Our proposed SPTNet demonstrates superior accuracy while obtaining mostly the best time efficiency.

Table 13: Time efficiency of different methods on ImageNet-100 and SSB. Bold values represent the best results.

| Method | ImageNet-100 | | | SSB | | |
|---|---|---|---|---|---|---|
| | Accuracy (All) | Training time (Sec) | Inference time (Sec) | Accuracy (All) | Training time (Sec) | Inference time (Sec) |
| GCD Vaze et al. (2022) | 74.1 | 803 | 2289 | 51.3 | 58 | 552 |
| SimGCD Wen et al. (2023) | 83.0 | 847 | **591** | 56.1 | 64 | **17** |
| PromptCAL Zhang et al. (2023) | 83.1 | 1817 | 893 | 55.1 | 492 | 103 |
| SPTNet (Ours) | **85.4** | **483** | 601 | **61.4** | **32** | **17** |

## J   THEORETICAL ANALYSIS ON OUR ALTERNATE TRAINING

To estimate the model parameters $\theta$, it is common to introduce the log-likelihood function $L(\theta) = \ln \mathcal{P}(X \mid \theta)$. This function quantifies the likelihood of the parameter $\theta$ given the data $X$. As the natural logarithm function, $\ln X$, is monotonically increasing, maximizing $\mathcal{P}(X \mid \theta)$ is equivalent to maximizing $L(\theta)$. In other words, maximizing the log-likelihood function $L(\theta)$ achieves the same objective.

The EM algorithm is an iterative procedure designed to maximize $L(\theta)$. Let $\theta_t$ denote the current estimate for $\theta$ after the $t$-th iteration. Our goal is to calculate an updated estimate $\theta$ that maximizes $L(\theta)$:

$$
\begin{aligned}
L(\theta) - L(\theta_t) &= \ln \mathcal{P}(X \mid \theta) - \ln \mathcal{P}(X \mid \theta_t) \\
&= \ln \left( \sum_{p^{1:n}} \mathcal{P}(X \mid p^{1:n}, \theta) \mathcal{P}(p^{1:n} \mid \theta) \right) - \ln \mathcal{P}(X \mid \theta_t) \\
&= \ln \left( \sum_{p^{1:n}} \mathcal{P}(X \mid p^{1:n}, \theta) \mathcal{P}(p^{1:n} \mid \theta) \cdot \frac{\mathcal{P}(p^{1:n} \mid X, \theta_t)}{\mathcal{P}(p^{1:n} \mid X, \theta_t)} \right) - \ln \mathcal{P}(X \mid \theta_t) \\
&= \ln \left( \sum_{p^{1:n}} \mathcal{P}(p^{1:n} \mid X, \theta_t) \frac{\mathcal{P}(X \mid p^{1:n}, \theta) \mathcal{P}(p^{1:n} \mid \theta)}{\mathcal{P}(p^{1:n} \mid X, \theta_t)} \right) - \ln \mathcal{P}(X \mid \theta_t) \qquad (5) \\
&\geq \sum_{p^{1:n}} \mathcal{P}(p^{1:n} \mid X, \theta_t) \ln \left( \frac{\mathcal{P}(X \mid p^{1:n}, \theta) \mathcal{P}(p^{1:n} \mid \theta)}{\mathcal{P}(p^{1:n} \mid X, \theta_t)} \right) - \ln \mathcal{P}(X \mid \theta_t) \\
&= \sum_{p^{1:n}} \mathcal{P}(p^{1:n} \mid X, \theta_t) \ln \left( \frac{\mathcal{P}(X \mid p^{1:n}, \theta) \mathcal{P}(p^{1:n} \mid \theta)}{\mathcal{P}(p^{1:n} \mid X, \theta_t) \mathcal{P}(X \mid \theta_t)} \right) \\
&\triangleq H(\theta \mid \theta_t).
\end{aligned}
$$

Let $l(\theta \mid \theta_t) = L(\theta_t) + H(\theta_t \mid \theta_t)$ which is bounded above by the likelihood function $L(\theta)$. Let $\theta = \theta_t$, we observe that:

$$
\begin{aligned}
l(\theta_t \mid \theta_t) &= L(\theta_t) + H(\theta_t \mid \theta_t) \\
&= L(\theta_t) + \sum_{p^{1:n}} \mathcal{P}(p^{1:n} \mid X, \theta_t) \ln \frac{\mathcal{P}(X \mid p^{1:n}, \theta_t) \mathcal{P}(p^{1:n} \mid \theta_t)}{\mathcal{P}(p^{1:n} \mid X, \theta_t) \mathcal{P}(X \mid \theta_t)} \\
&\neq L(\theta_t) + \sum_{p^{1:n}} \mathcal{P}(p^{1:n} \mid X, \theta_t) \ln \frac{\mathcal{P}(X, p^{1:n} \mid \theta_t)}{\mathcal{P}(X, p^{1:n} \mid \theta_t)} \qquad (6) \\
&= L(\theta_t) + \sum_{p^{1:n}} \mathcal{P}(p^{1:n} \mid X, \theta_t) \ln 1 \\
&= L(\theta_t).
\end{aligned}
$$

Note that in general, $\mathcal{P}(X \mid p^{1:n}, \theta_t) \mathcal{P}(p^{1:n} \mid \theta_t)$ cannot be equal to $\mathcal{P}(X, p^{1:n} \mid \theta_t)$ since $p^{1:n}$ is conditioned on both $X$ and $\theta$. Consequently, the factorized form does not equal the joint distribution. As a result, for $\theta = \theta_t$, the functions $l(\theta \mid \theta_t)$ and $L(\theta)$ are not equal.

Our objective is to find the $\theta$ that maximizes the function $L(\theta)$. While $l(\theta|\theta_t)$ and $L(\theta)$ may not be equal for the current estimate $\theta = \theta_t$, it still holds that $l(\theta|\theta_t)$ is bounded by $L(\theta)$. Therefore, increasing $l(\theta|\theta_t)$ will also increase $L(\theta)$. To achieve the greatest increase in $L(\theta)$, the EM algorithm selects an updated $\theta_{t+1}$ that maximizes $l(\theta|\theta_t)$:

$$
\begin{aligned}
\theta_{t+1} &= \arg\max_{\theta} \{ l(\theta \mid \theta_t) \} \\
&= \arg\max_{\theta} \left\{ L(\theta_t) + \sum_{p^{1:n}} \mathcal{P}(p^{1:n} \mid X, \theta_t) \ln \frac{\mathcal{P}(X \mid p^{1:n}, \theta) \mathcal{P}(p^{1:n} \mid \theta)}{\mathcal{P}(X \mid \theta_t) \mathcal{P}(p^{1:n} \mid X, \theta_t)} \right\} \qquad (7)
\end{aligned}
$$

Ignoring terms which are constant w.r.t. $\theta$, the equation can be further deduced:

$$
\begin{aligned}
\theta_{t+1} &= \arg\max_{\theta} \left\{ \sum_{p^{1:n}} \mathcal{P}\left(p^{1:n} \mid X, \theta_t\right) \ln \mathcal{P}(X \mid p^{1:n}, \theta) \mathcal{P}(p^{1:n} \mid \theta) \right\} \\
&= \arg\max_{\theta} \left\{ \sum_{p^{1:n}} \mathcal{P}\left(p^{1:n} \mid X, \theta_t\right) \ln \frac{\mathcal{P}(X, p^{1:n}, \theta)}{\mathcal{P}(p^{1:n}, \theta)} \frac{\mathcal{P}(p^{1:n}, \theta)}{\mathcal{P}(\theta)} \right\} \\
&= \arg\max_{\theta} \left\{ \sum_{p^{1:n}} \mathcal{P}\left(p^{1:n} \mid X, \theta_t\right) \ln \mathcal{P}(X, p^{1:n} \mid \theta) \right\} \\
&= \arg\max_{\theta} \left\{ \mathrm{E}_{p^{1:n}\mid X, \theta_t} \left\{ \ln \mathcal{P}(X, p^{1:n} \mid \theta) \right\} \right\}.
\end{aligned}
\tag{8}
$$

The alternate training algorithm thus consists of iterating (1) E-step: Determine the conditional expectation $\mathrm{E}_{p^{1:n}\mid X, \theta_t}\{\ln \mathcal{P}(X, p^{1:n} \mid \theta)\}$. (2) M-step: Maximize this expression with respect to $\theta$. It is evident that end-to-end training for maximizing $L(\theta)$ is not equivalent to two-stage learning $l(\theta|\theta_t)$ in the converged state, as verified in Eq. (6). Another advantage of two-stage learning is that it provides a framework for better estimation for both model and data parameters. This is further supported by the evidence presented in Fig. 8, where end-to-end training leads to sub-optimal solutions for $p^{1:n}$.

## K  MORE VISUALIZATION AND ANALYSIS OF ATTENTION MAPS

We visualize attention maps from different heads in the last layer of the ViT backbone for multiple datasets. The query position is selected either as the *CLS* token (in Fig. 11, Fig. 12, Fig. 13) or the local region on the edge of the foreground object (in Fig. 14, Fig. 15, Fig. 16, Fig. 17). We show the top 60% most attended patches in red for different attention heads. We observe that SPTNet-P and SPTNet can attend to more salient object regions, likely due to their ability to learn local invariance. Besides, SPTNet-P and SPTNet cover more diverse regions of the salient object regardless of query positions, illustrating more diverse attention patterns across heads. A similar phenomenon can be found in Stanford Cars and FGVC-Aircraft in Fig. 11 and Fig. 12 for the *CLS* token and Fig. 15 and Fig. 16 for the edge position, as well as in the generic dataset (*e.g.*, ImageNet) in Fig. 13 and Fig. 17.

We also investigate the impact of our proposed SPT by separating the prompt and backbone from SPTNet-P. We remove the prompt component from a well-trained SPTNet-P and visualize the attention maps by feeding raw images to the backbone only, referred to as SPTNet-P (w/o prompt)'. Upon comparing the attention maps with and without patch-wise prompts, we observe that SPTNet-P with prompts (*i.e.*, the $3^{rd}$ column) exhibits clearer attention on foreground objects compared to SPTNet-P without prompts (*i.e.*, the $2^{nd}$ column). This indicates that the learned prompts help elicit critical features for recognition. Additionally, when considering a generic dataset like ImageNet-100, we notice that there is no significant difference between the attention maps of models with and without prompts, resulting in an inferior performance boost compared to the fine-grained datasets.

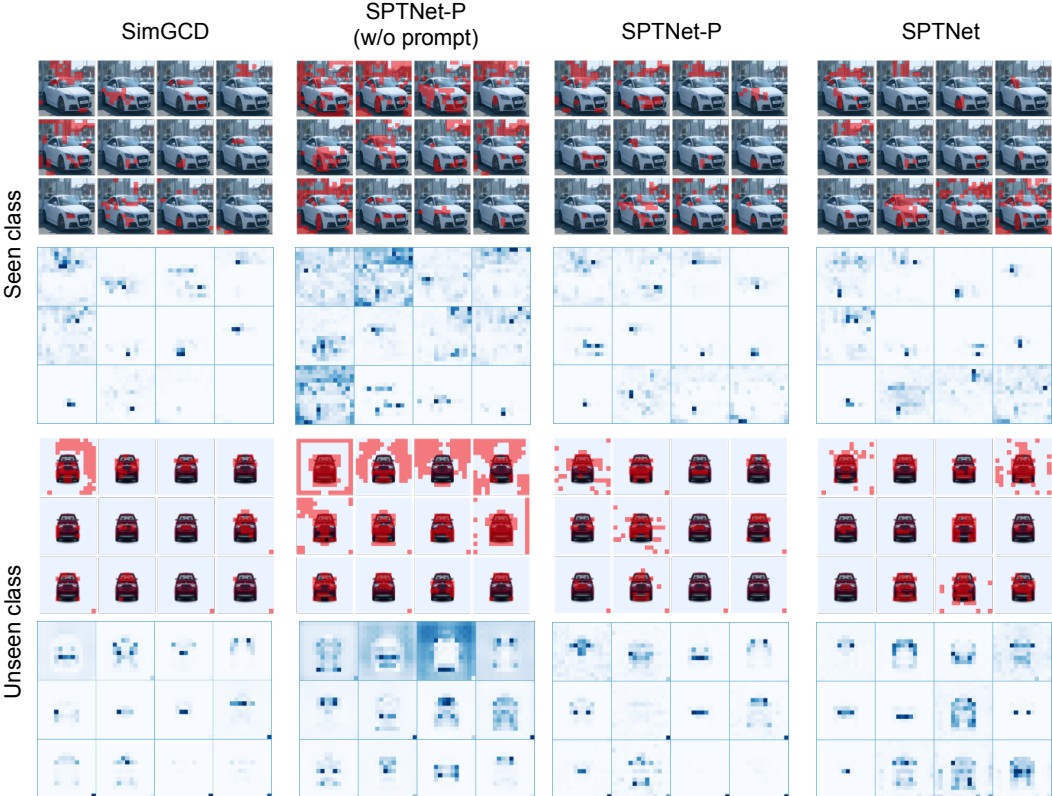

Figure 11: Attention visualization on Stanford Cars, for 12 different attention heads in the last layer of the ViT backbone, by querying the *CLS* token.

To explore the difference of performance boost between generic and fine-grained datasets, we transfer a pre-trained model from a fine-grained dataset to a generic one in Fig. 22. Since the overlap between ImageNet-100 and fine-grained datasets only contains various types of birds, we select some *bird samples from ImageNet-100* as inputs and visualize the attention map of two models: (1) trained on ImageNet-100 and (2) trained on CUB. The model trained on CUB appears to focus more on local regions, while the one trained on ImageNet-100 pays more attention to the entire object. Furthermore, based on the quantitative comparison presented in Section 4.3 and Appendix C, we observe that SPTNet-P outperforms SPTNet on the ImageNet-100 dataset but performs worse than SPTNet on CUB. Additionally, as depicted in Figure 22, we can observe that when more diverse attention is focused on different regions of the object, it corresponds to improved performance. This indicates that

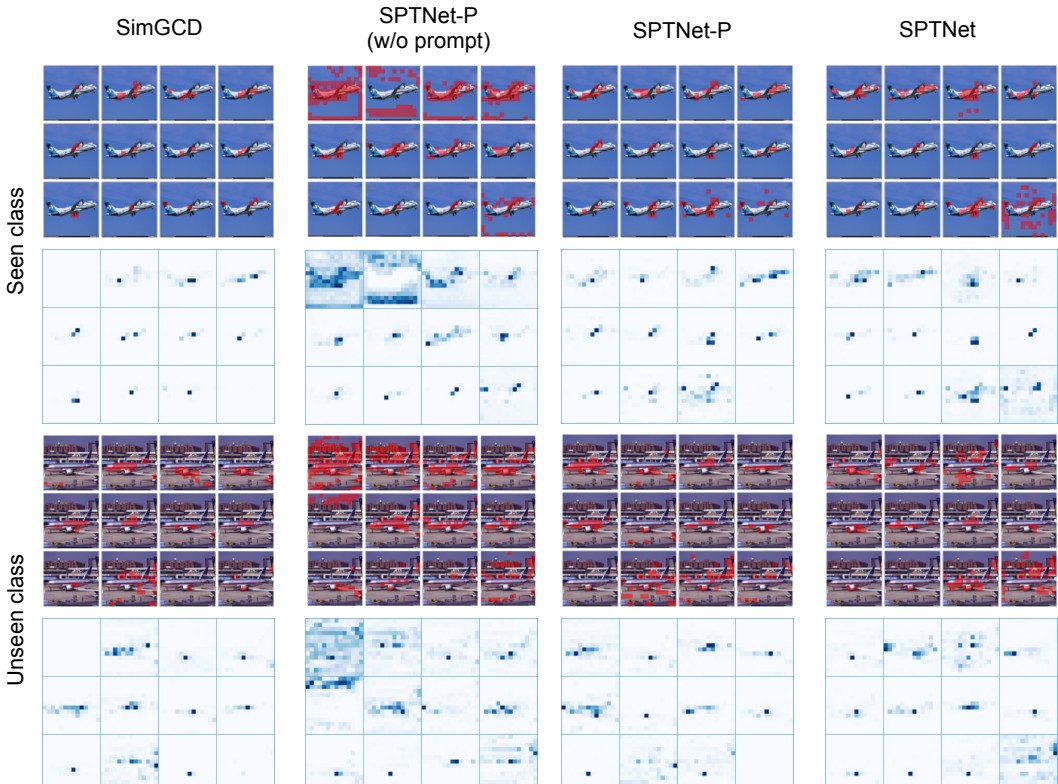

Figure 12: Attention visualization on FGVC-Aircraft, for 12 different attention heads in the last layer of the ViT backbone, by querying the *CLS* token.

the ability of the model to concentrate attention on various object regions is beneficial for achieving better results. For instance, when comparing 'SPTNet-P' and 'SPTNet' trained on the ImageNet-100 dataset, we observe that 'SPTNet-P' exhibits a higher concentration on the objects compared to 'SPTNet.' This observation aligns with the qualitative comparison, indicating that 'SPTNet-P" performs better than 'SPTNet' on ImageNet-100. Similarly, when considering 'SPTNet-P (from CUB)' and 'SPTNet (from CUB)' trained on the CUB dataset, we notice that 'SPTNet' demonstrates a stronger focus on the objects compared to 'SPTNet-P'. This observation is consistent with the qualitative comparison, suggesting that 'SPTNet' outperforms 'SPTNet-P' on CUB.

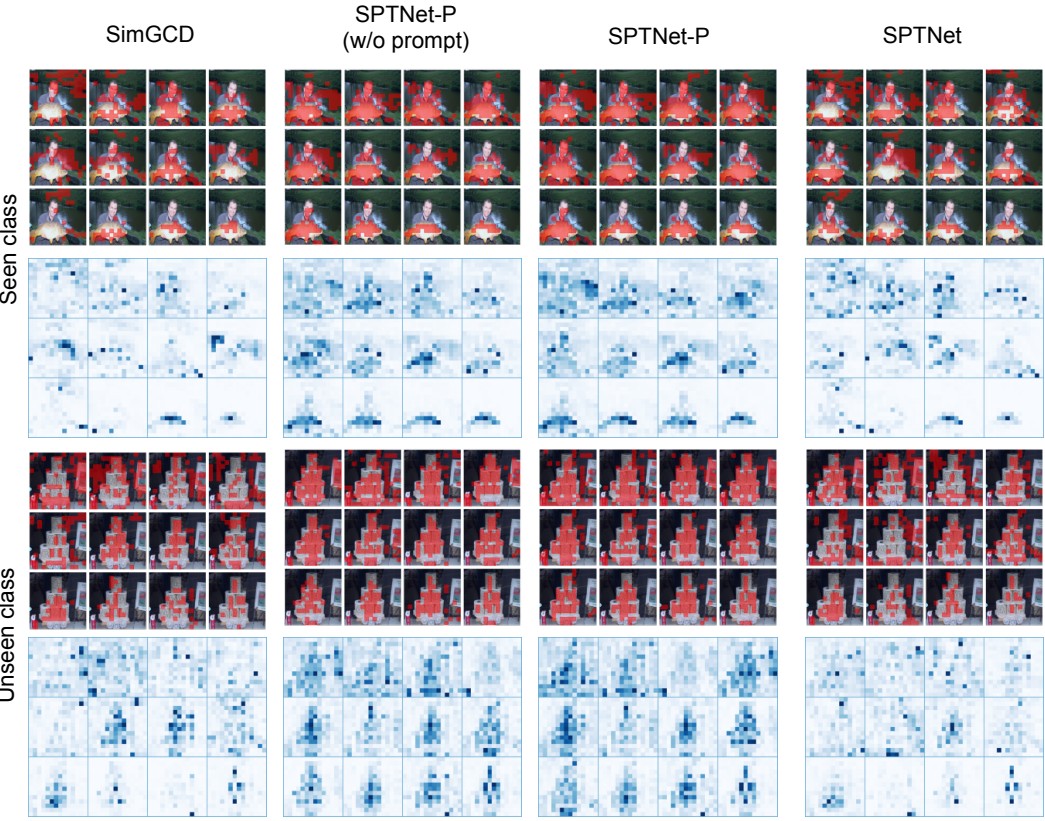

Figure 13: Attention visualization on ImageNet-100, for 12 different attention heads in the last layer of the ViT backbone, by querying the *CLS* token.

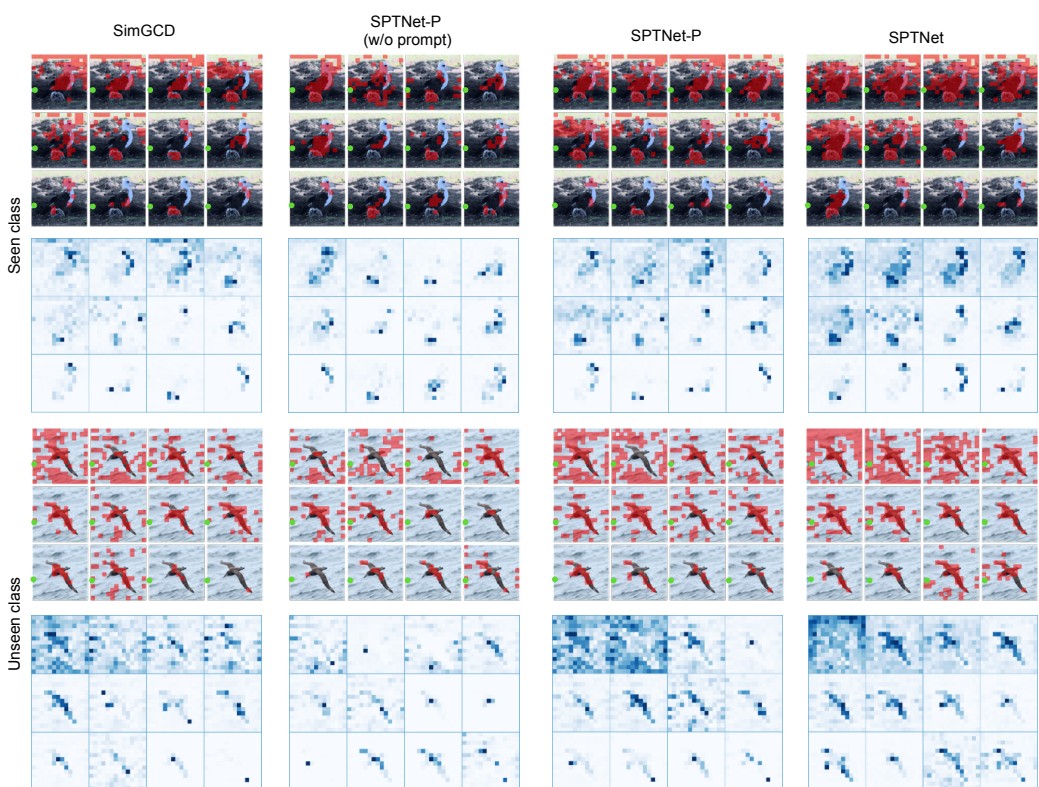

Figure 14: Attention visualization on CUB, for 12 different attention heads in the last layer of the ViT backbone, by querying the point marked as green.

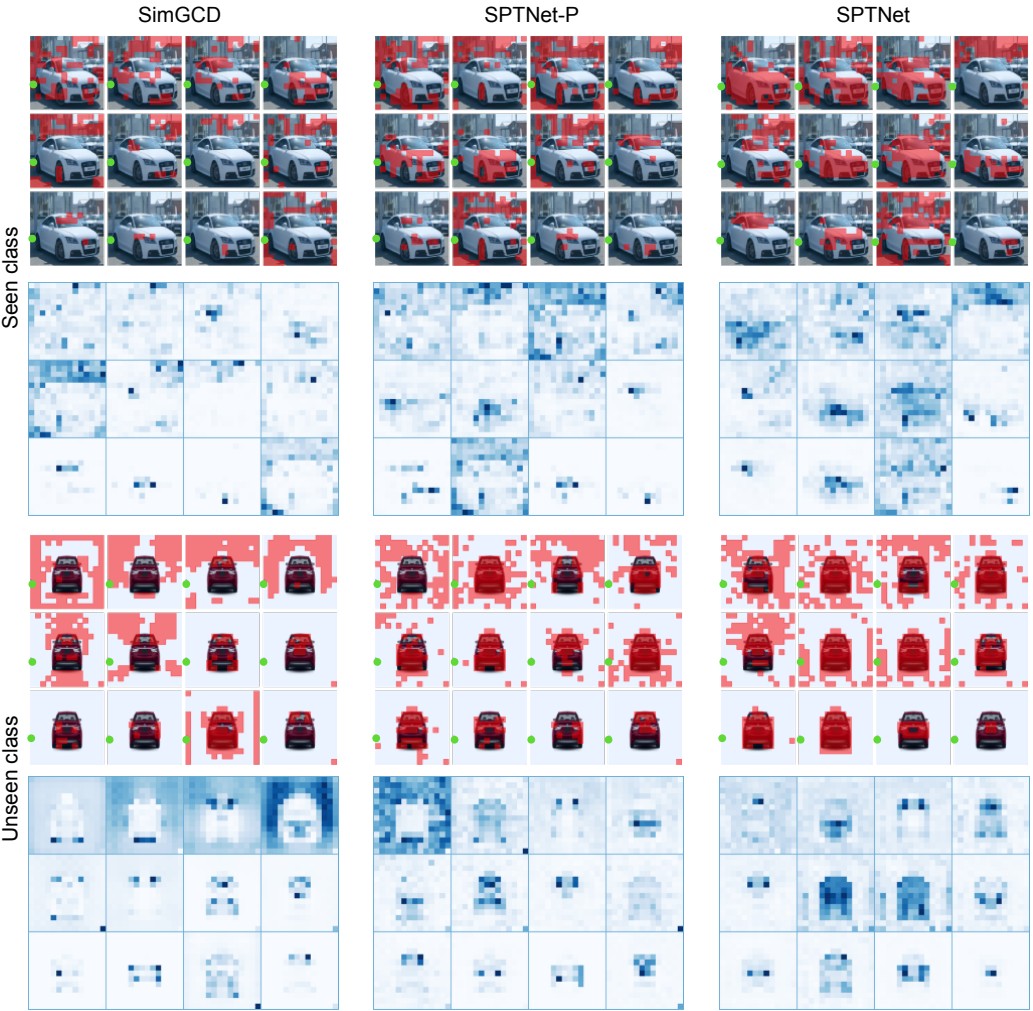

Figure 15: Attention visualization on Stanford Cars, for 12 different attention heads in the last layer of the ViT backbone, by querying the point marked as green.

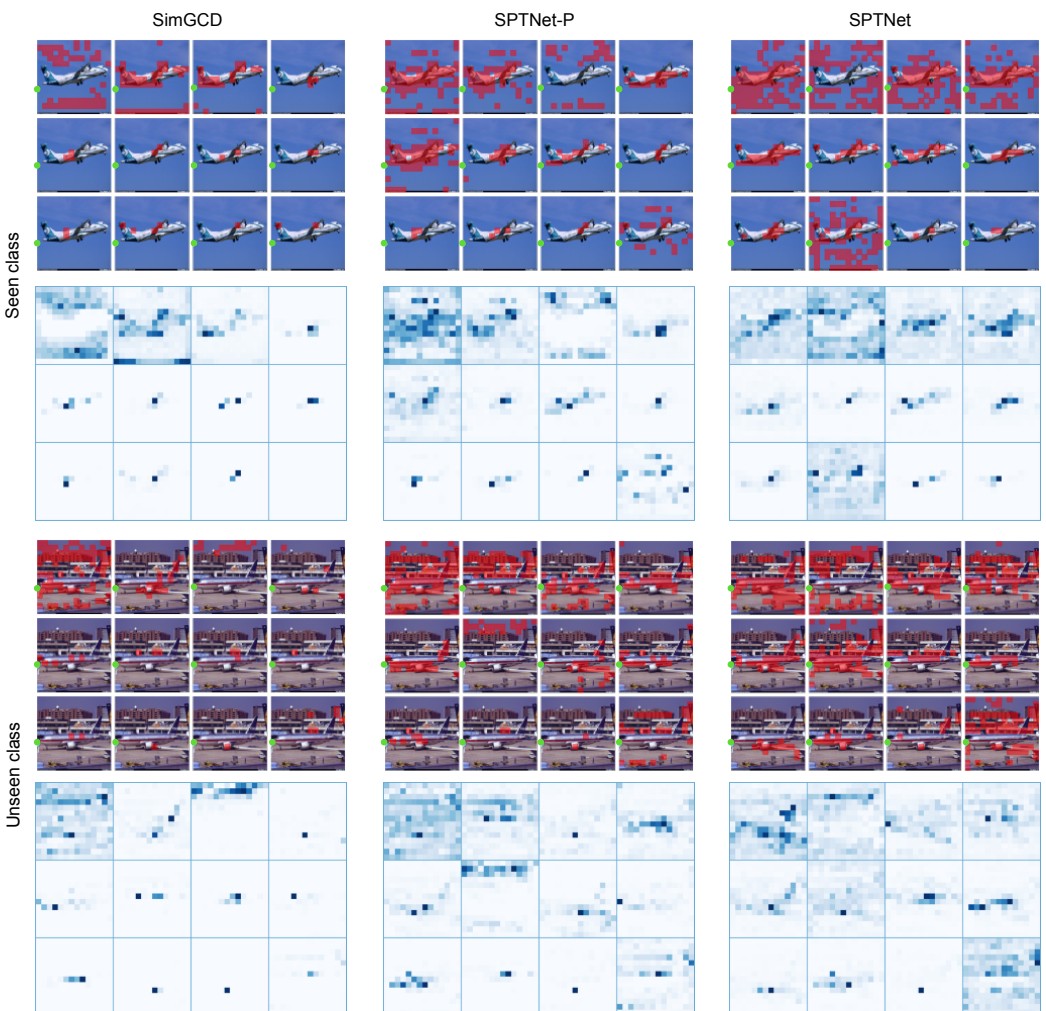

Figure 16: Attention visualization on FGVC-Aircraft, for 12 different attention heads in the last layer of the ViT backbone, by querying the point marked as green.

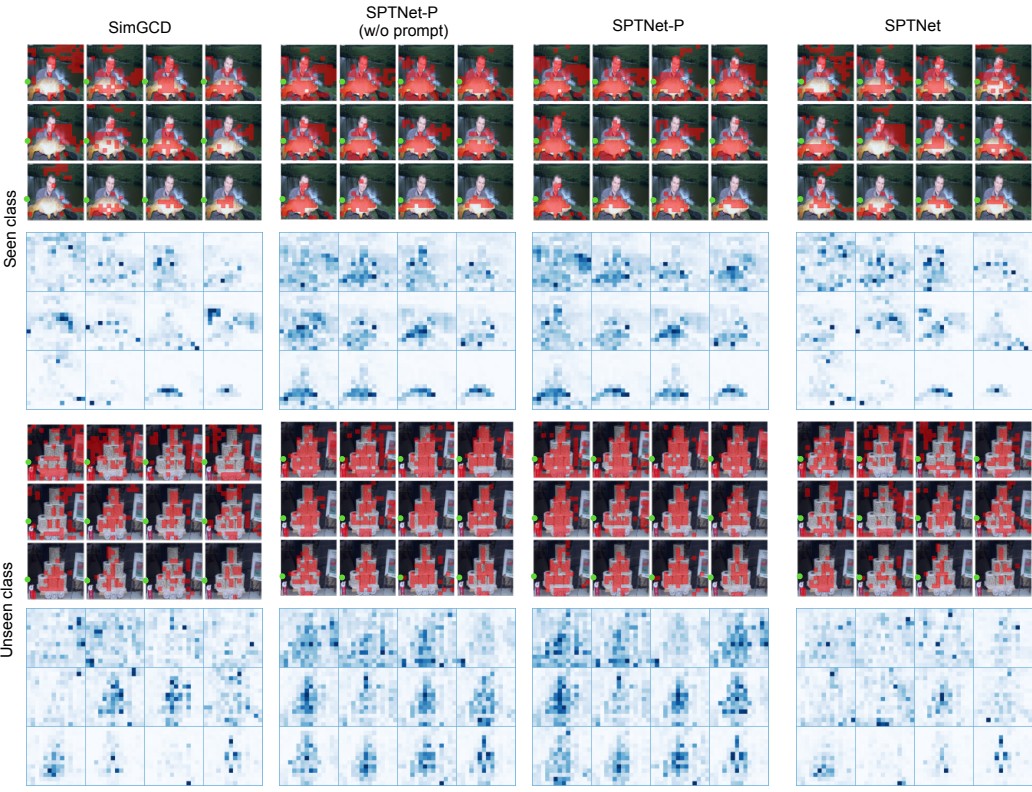

Figure 17: Attention visualization on ImageNet-100, for 12 different attention heads in the last layer of the ViT backbone, by querying the point marked as green.

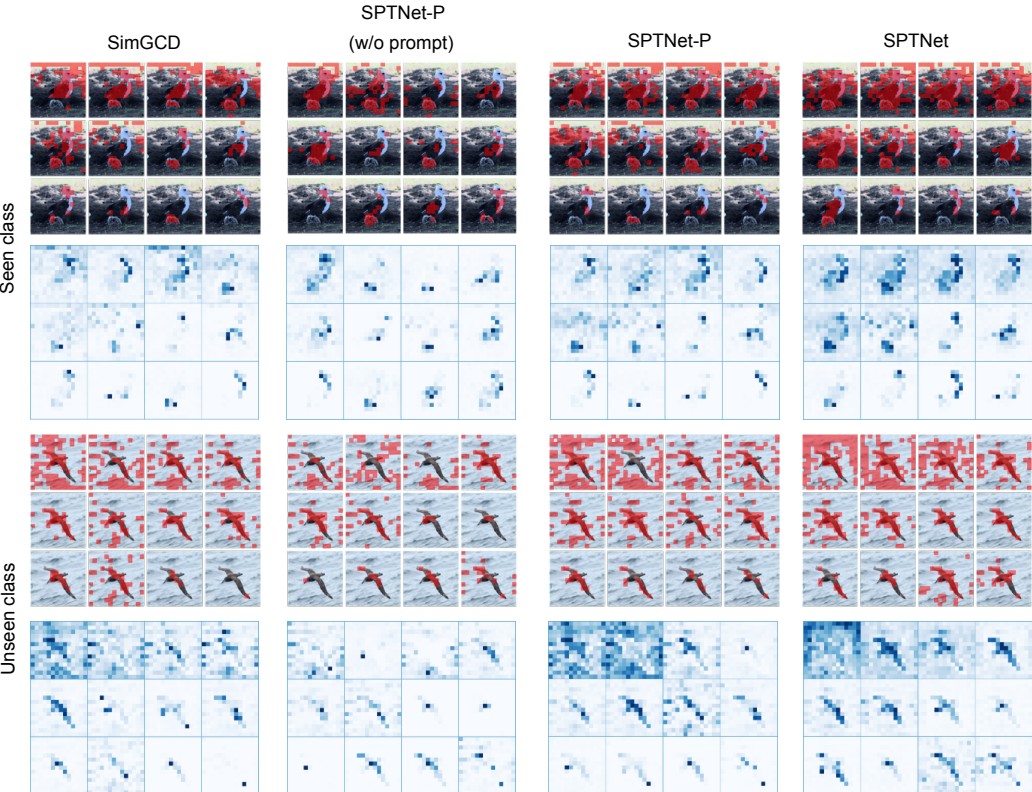

Figure 18: Attention visualization on CUB, for 12 different attention heads in the last layer of the ViT backbone, by querying the *CLS* token. SPTNet-P and SPTNet can automatically identify salient objects, likely due to their ability to learn local invariance. Upon comparing the attention maps with and without patch-wise prompts, we observe that SPTNet with prompts (*i.e.*, the $3^{rd}$ column) exhibits more concentrated attention on the salient object compared to SPTNet without prompts (*i.e.*, the $2^{nd}$ column). This indicates that our learned prompts help elicit critical features for recognition.

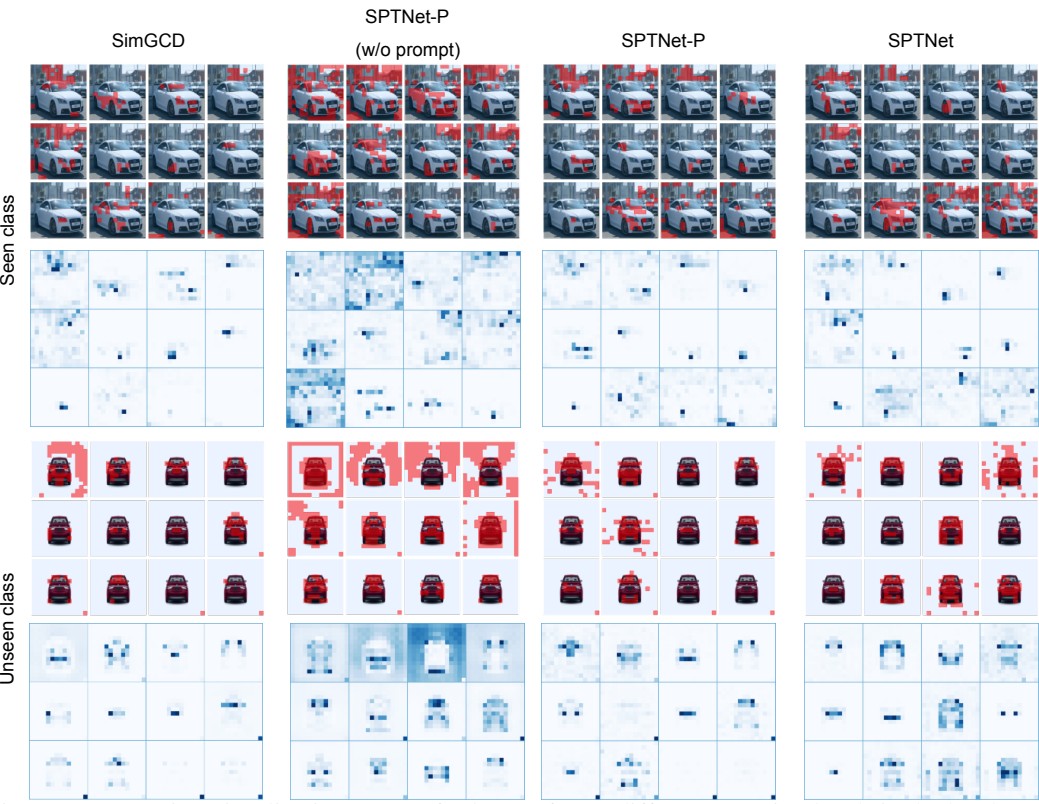

Figure 19: Attention visualization on Stanford Cars, for 12 different attention heads in the last layer of the ViT backbone, by querying the *CLS* token. SPTNet-P and SPTNet can automatically identify salient objects, likely due to their ability to learn local invariance. Upon comparing the attention maps with and without patch-wise prompts, we observe that SPTNet with prompts (*i.e.*, the $3^{rd}$ column) exhibits more concentrated attention on the salient object compared to SPTNet without prompts (*i.e.*, the $2^{nd}$ column). This indicates that our learned prompts help elicit critical features for recognition.

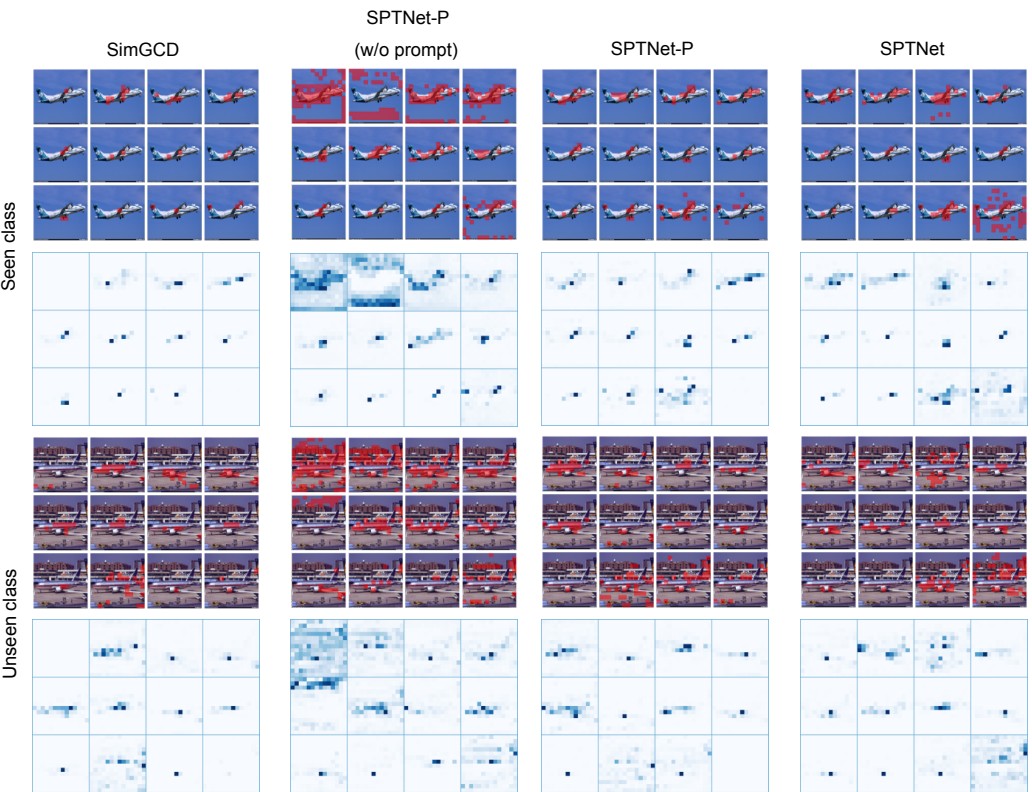

Figure 20: Attention visualization on FGVC-Aircraft, for 12 different attention heads in the last layer of the ViT backbone, by querying the *CLS* token. SPTNet-P and SPTNet can automatically identify salient objects, likely due to their ability to learn local invariance. Upon comparing the attention maps with and without patch-wise prompts, we observe that SPTNet with prompts (*i.e.*, the $3^{rd}$ column) exhibits more concentrated attention on the salient object compared to SPTNet without prompts (*i.e.*, the $2^{nd}$ column). This indicates that our learned prompts help elicit critical features for recognition.

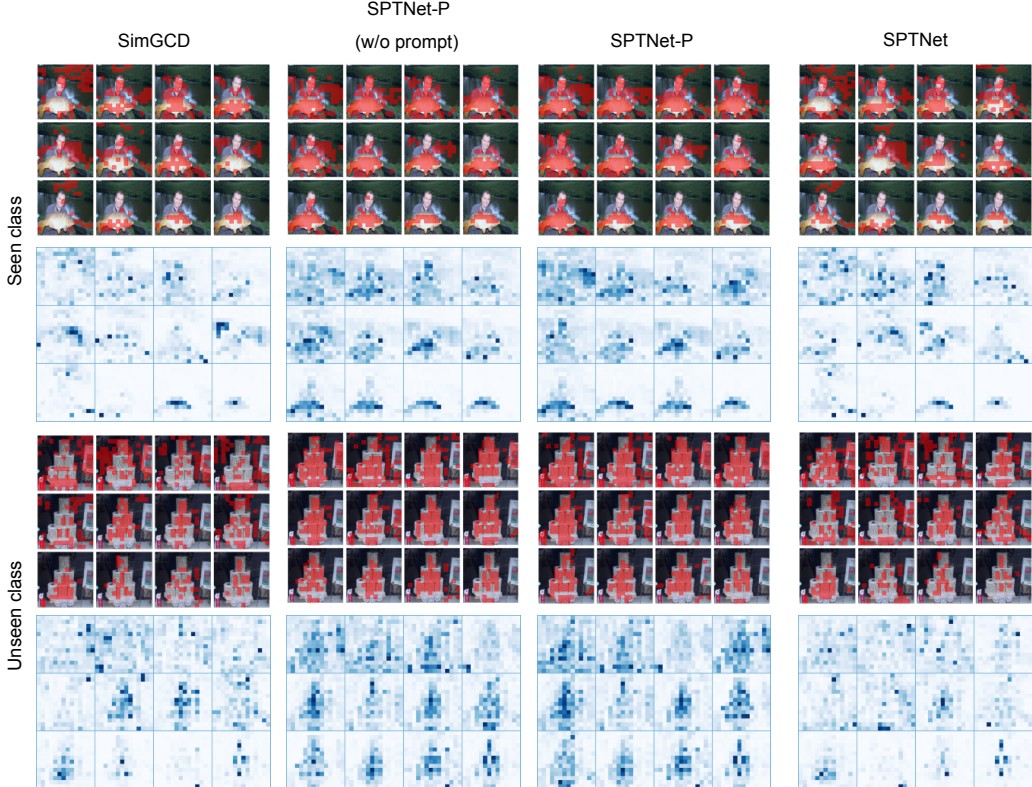

Figure 21: Attention visualization on ImageNet-100, for 12 different attention heads in the last layer of the ViT backbone, by querying the *CLS* token. SPTNet-P and SPTNet can automatically identify salient objects, likely due to their ability to learn local invariance. Upon comparing the attention maps with and without patch-wise prompts, we observe that SPTNet with prompts (*i.e.*, the $3^{rd}$ column) exhibits more concentrated attention on the salient object compared to SPTNet without prompts (*i.e.*, the $2^{nd}$ column). This indicates that our learned prompts help elicit critical features for recognition.

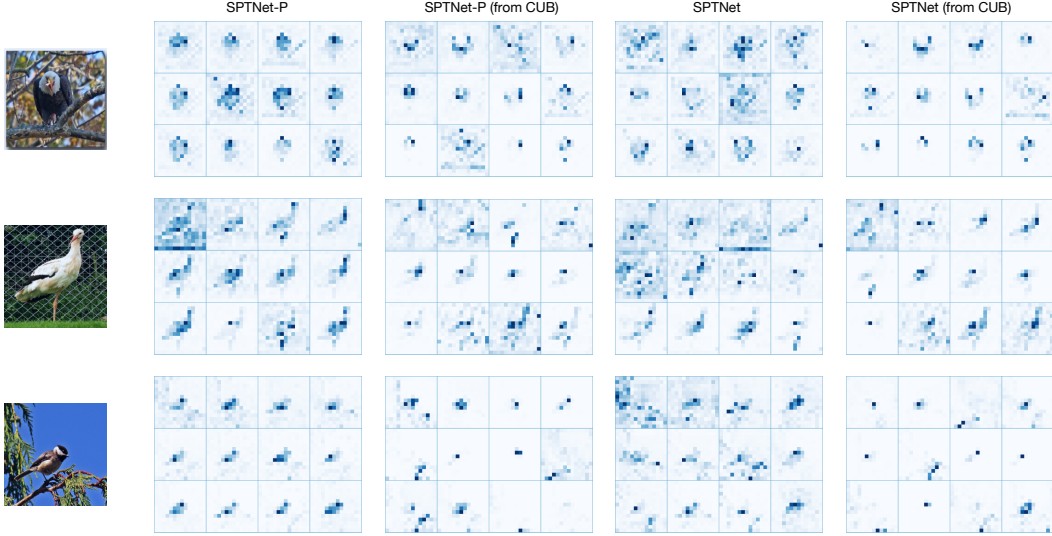

Figure 22: Attention visualization for models trained on ImageNet-100 and CUB by applying them to bird images from ImageNet-100. The models trained on CUB are marked as '(from CUB)'.

## L  BROADER IMPACTS

Our study is among the efforts to extend the capability of AI systems from the closed world to the open world. Particularly, it will play a positive role in fostering next-generation AI systems with the capability of categorizing and organizing open-world data automatically. However, our method still has several limitations. First, though we have achieved encouraging results on the public datasets, the interpretability still needs improvement, as the underlying principles of how the decisions are made by the systems remain not crystal clear. Second, the cross-domain robustness is not satisfactory, as can be seen from the results on the setting of GCD with domain shifts, though our method has achieved the best overall results and new class discovery results, the performance still has significant room to improve. Additionally, in the vanilla GCD setting, methods typically rely on a pre-trained model (*e.g.*, DINO) as a feature extractor, which may inherit its drawbacks (*e.g.*, discrimination and privacy issues).

