# OpenReview forum: "SPTNet: An Efficient Alternative Framework for Generalized Category Discovery with Spatial Prompt Tuning"
_ICLR.cc/2024/Conference — ICLR 2024 poster_

### Official Review · Reviewer_E8qj · 2023-10-27

**Soundness:** 3 good
**Presentation:** 3 good
**Contribution:** 4 excellent
**Rating:** 8
**Confidence:** 5

**Summary:**

- Generalized Category Discovery (GCD) is the problem of leveraging information from known classes in labeled data to automatically identify known and unknown classes in unlabeled data. Authors propose a two stage aporoach, named SPTNet for GCD.
- SPTNet iteratively optimizes "model" parameters of large self-supervised networks and "data" parameters (i.e. prompt tuning methods). The former adapts the model to the data while the latter adapts the data to improve the model's capability of identifying categories.
- Authors propose a novel Spatial Prompt Tuning (SPT) method that enables the model to focus on object parts an show remarkable improvements across several GCD benchmarks with very few additional trainable parameters.

**Strengths:**

- The idea of optimizing the model and data parameters to improve GCD has some merit especially for discovery in fine-grained datasets.
- The experiments section supports most of the claims (except a few discussed in weaknesses) made in the paper and show the merit of the proposed approach.

**Weaknesses:**

**Interpretability of SPT**:
- Authors claim that SPT enables the model to focus on parts of objects, but the way it is designed, there are learnable parameters around each patch. Are the learned parameters, after convergence, sparse in nature? Are there more non-zero values around discriminative regions of objects and zero around patches belonging to background? The SPT setup in its current form is not very interpretable and the experiments certainly do not validate the claim that SPT is better than Bahng et al. because it enables the model to focus on object parts.
Can this claim be validated/negated if, instead of around a patch, SPT is applied as learnable horizontal or vertical stripes in the image (maintaining the same number of parameters as the original SPTNet). If these experiments achieve the same performance, then the claim made by the authors is not true. I would like to hear the authors' thoughts on this.
**Ablation experiments**:
- Table-4 is not presented efficiently in my opinion and needs more attention. From rows 6 and 7, without Global prompting, SPT-S is better than SPT-P. But there is no experiment which uses Global + SPT-S with alternate training in the table.
- I recommend adding one component at a time to the baseline, makes the table more readable than its current form. I believe most of the experiments are in there and all one would need is to rearrange the rows accordingly.
- Also the accompanying text to Table-4 has some mistakes which make it harder to read the table. For example. Rows 5,6 compare the effect of alternate training. But in the text, authors explain that these two rows show the benefit of global prompting. Kindly make the table and text consistent and readable to improve reading experience.
**Alternate training**:
- In Fig. 3(a), for each k, are epochs adjusted accordingly? I believe this is important because authors report that with smaller k, the model underfits. But what about the experiment of training the model parameters to convergence, followed by training the SPT to convergence (k=1). This experiment is crucial to understand why the alternate training is required. Please provide the details of this experiment.
**Qualitative results**:
Authors show attention maps in Fig. 4b and claim that with SPT, the heads cover the object. I do not see a difference between visualizations of SimGCD and SPTNet to be honest. Instead of showing 4 examples with all the heads, I recommend authors to be more specific and show exactly (by highlighting the region) what they want the readers to focus on. In its current form Fig. 4 does not add anything new to the discussion and can be removed entirely to make space for more important experiments (suggested above).

**Questions:**

**Suggestions**:
- The writing of the paper can be improved. Few sections (ablation, Section 3.2) need improvement.
- Font of text in Fig.1 font is too small and I recommend increasing that.
- Difference between SPTNet and SPTNet-P is not clear in the main paper. Readers have to look to supplementary to get clarity. Since this is an important part of the contribution, its better authors move Fig. 5 to the main paper. Also provide an example of computing total number of parameters of SPT in the supplementary.

**Questions**:
- What happens when you train Bahng et. al's Global prompts by increasing the number of parameters to match SPT? I understand its not a whole lot of parameters but how much would that change the performance of using Global prompts?
- In the paragraph below **Stage 2: Fix p_{1:n}...**, authors present that they use the spatial prompts as augmentations. But by using learned prompts as augmentation, you are asking the network to be invariant to it, how does this help? The intuition behind using this and why would it improve the performance? How much does it improve the performance of contrastive loss? If that is a significant improvement, then that would be a good result for self-supervised literature.

**Please address all the concerns and questions raised above for me to improve my ratings**

**Details Of Ethics Concerns:**

I do not foresee any immediate ethical concerns with this work.

---

> ### Author Response · Authors · 2023-11-21
> **Response to Reviewer #E8qj**
>
> > Authors claim that SPT enables the model to focus on parts of objects, but the way it is designed, there are learnable parameters around each patch. Are the learned parameters, after convergence, sparse in nature? Are there more non-zero values around discriminative regions of objects and zero around patches belonging to background? The SPT setup in its current form is not very interpretable and the experiments certainly do not validate the claim that SPT is better than Bahng et al. because it enables the model to focus on object parts. Can this claim be validated/negated if, instead of around a patch, SPT is applied as learnable horizontal or vertical stripes in the image (maintaining the same number of parameters as the original SPTNet).
>
>
> Thanks for the comments. To validate the sparsity of our learned prompts, we visualize and analyze the learned prompts in Appendix E. It can be seen that after learning, the majority of learned prompts are activated, so the learned prompts are not sparse.
>
> As discussed in General Response -Q1, SPT serves as a learned data augmentation technique while data augmentation is very important for contrastive learning in GCD. Therefore, more activated prompt parameters means a stronger augmentation; while sparsely activated prompt parameters means a weaker augmentation, which is unlikely to enhance the representation learning much.
> The effectiveness of the learned spatial prompts is also validated by the attention maps, as shown in Fig. 4(b), which demonstrate more diverse attention maps across different heads (top right subfigure).
>
>
> We follow the suggestion to apply learnable horizontal and vertical stripes separately and the results are shown in Table S2, with the "width" of the prompt, $s$, set to 2, leading to 0.2 "data" parameters. Note that the $s$ for SPT-Net is set to 1, leading to 0.1M learnable `data’ parameters. If we set $s$ for the stripe veraints, there will be 0.03M learnable parameters. Hence, we choose $s=2$ in this experiment for better capacity of the stripe variants. As can be seen from Table S2, Despite having slightly more parameters than SPTNet, these two variantes are still outperformed by SPTNet, indicating that the improvement is due to the design rather than parameters.
>
> Table S2: Evaluation on SSB. Bold values represent the best results.
> |                             | # Trainable parameters(#model parameters + #data parameters) | All      | Old      | New      |
> |-----------------------------|--------------------------------------------------------------|----------|----------|----------|
> | SimGCD                      | **14.1M+0M**                                                 | 56.1     | 65.5     | 51.5     |
> | SPTNet (horizontal)         | 14.1M+0.102M                                                   | 59.0     | 68.1     | 54.3     |
> | SPTNet (vertical)           | 14.1M+0.102M                                                   | 59.4     | 68.2     | 54.8     |
> | SPTNet (alternate training) | 14.1M+0.100M                                                   | **61.4** | **69.9** | **57.5** |
>
> > Table-4 is not presented efficiently in my opinion and needs more attention. From rows 6 and 7, without Global prompting, SPT-S is better than SPT-P. But there is no experiment which uses Global + SPT-S with alternate training in the table.
>
>
> Thanks for the suggestion. We added the experiment for Global + SPT-S and rearranged rows in Table-4 to show the influence of different components step by step in the revised paper. We create Table S3 here by selecting relevant results from Table-4 in the main paper for convenience. The comparison of SPT-P and SPT-S with global prompting reveals that SPT-P outperforms SPT-S, indicating that  SPT-P is also more effective than SPT-S when coupled with the global prompting and alternative training.
>
> Table S3: Evaluation on SSB. Bold values represent the best results.
> |                      | All      | Old      | New      |
> |----------------------|----------|----------|----------|
> | SimGCD               | 56.1     | 65.5     | 51.5     |
> | Global + SPT-S+Alter | 60.9     | 69.0     | 57.3     |
> | Global + SPT-P+Alter | **61.4** | **69.9** | **57.5** |
>
> > In Fig. 3(a), for each k, are epochs adjusted accordingly? I believe this is important because authors report that with smaller k, the model underfits. But what about the experiment of training the model parameters to convergence, followed by training the SPT to convergence (k=1). This experiment is crucial to understand why the alternate training is required. Please provide the details of this experiment.
>
>
> Thanks for the suggestions. We have provided the the suggested experiments and analysis in General Response - Q3.

---

> > ### Author Response · Authors · 2023-11-21
> >
> > > Difference between SPTNet and SPTNet-P is not clear in the main paper. Readers have to look to supplementary to get clarity. Since this is an important part of the contribution, its better authors move Fig. 5 to the main paper. Also provide an example of computing total number of parameters of SPT in the supplementary.
> >
> >
> > Thanks for your suggestion. We added an example for computing the total number of parameters of SPT in Appendix B and would relocate Fig. 5 from the appendix to the main paper in our final manuscript for better clarity.
> >
> >
> > > What happens when you train Bahng et. al's Global prompts by increasing the number of parameters to match SPT? I understand its not a whole lot of parameters but how much would that change the performance of using Global prompts?
> >
> >
> > Thanks for the suggestion. Following the suggestion, we tested global prompts alone with alternate training using a prompt `width’  $s^{+}=50$. In this case, the total number of parameters is 0.1M, which is equal to SPT. The results are presented in Table S4. Comparing the $3^{rd}$ and $4^{th}$ rows, we observe that global prompts alone with alternate training do not outperform SPTNet despite having the same parameters. Additionally, global prompts perform better when $s^{+}=30$ (the default setting in our paper) than $s^{+}=50$.
> >
> > Table S4: Evaluation on SSB. Bold values represent the best results.
> > |                            | # Trainable parameters(#model parameters + #data parameters) | All      | Old      | New      |
> > |----------------------------|--------------------------------------------------------------|----------|----------|----------|
> > | SimGCD                     | **14.1M+0M**                                                 | 56.1     | 65.5     | 51.5     |
> > | Global ($s^{+}=30$) +Alter | 14.1M+0.07M                                                  | 57.8     | 66.3     | 53.8     |
> > | Global ($s^{+}=50$) +Alter | 14.1M+0.1M                                                   | 55.6     | 63.8     | 51.1     |
> > | SPTNet                     | 14.1M+0.1M                                                   | **61.4** | **69.9** | **57.5** |
> >
> > > In the paragraph below Stage 2: Fix p_{1:n}..., authors present that they use the spatial prompts as augmentations. But by using learned prompts as augmentation, you are asking the network to be invariant to it, how does this help? The intuition behind using this and why would it improve the performance? How much does it improve the performance of contrastive loss? If that is a significant improvement, then that would be a good result for self-supervised literature.
> >
> >
> > Good point! In our approach, we utilize learned prompts as augmentations in Stage 2. The use of learned prompts as augmentations is beneficial because it tailors the augmentation to the specific target setting, including the union dataset, architecture, base augmentations, and other factors. This tailored augmentation helps to improve performance by providing additional variations and challenges for the network to learn from. The idea of using learned prompts as augmentations may indeed be useful in the SSL literature. Our initial experiments did not show obvious gains with the method - we think this is because SSL augmentations are already very well tuned for the target setting. We suggest leaving an in depth investigation into this point to future work.
> >
> >
> > > Issues of typos, formatting and writing
> >
> >
> > We have made several improvements to enhance the clarity and consistency of our work. Specifically, we increased the text font in Fig. 1 as suggested. Additionally, we revised sections including ablation and Section 3.2 to improve readability and coherence and will further polish later. Furthermore, we simplified Fig. 4b and highlighted interesting regions for better illustration purposes. Lastly, we rearranged the formatting of Table 4 by adding different components step by step.

---

> > > ### Comment · Reviewer_E8qj · 2023-11-22
> > > **Response to authors' comments**
> > >
> > > I thank the reviewer for the detailed responses and additional experiments.
> > > - I believe the authors' point of SPT being better than Global prompt because it can focus on object parts was not addressed sufficiently. I recommend authors to support that statement with additional analysis or remove that claim from the paper.
> > > - Empirically authors show that the learnt "data" parameters work better when designed in a grid like fashion than using either horizontal or vertical stripes. I am not sure I fully understand why this is the case. I strongly suggest the authors perform additional analysis or atleast discuss (with supporting analysis) why this works better than other alternate strategies.
> > >
> > > I am fully satisfied with the authors' rebuttal and would like to improve my rating.

---

### Official Review · Reviewer_Pson · 2023-11-08

**Soundness:** 2 fair
**Presentation:** 3 good
**Contribution:** 4 excellent
**Rating:** 6
**Confidence:** 4

**Summary:**

The paper addresses the Generalized Category Discovery (GCD) problem, which involves training on labeled images from seen classes to classify both seen and unseen class images. The authors propose an alternative training method for prompts and model parameters, specifically introducing a spatial prompt tuning method that adds image prompts on a patch-wise basis. Their approach achieves a significant performance improvement of about 10% over existing methods using fewer parameters.

**Strengths:**

- The method proposed is both simple and highly effective, appearing easy to implement.

- Overall, except for the lack of method reasoning, the text is readable and has high-quality writing.

- Comprehensive evaluations across various datasets and against state-of-the-art methods are conducted, with thorough ablation studies and analyses supporting the proposed method.

**Weaknesses:**

- The paper's main weakness lies in the lack of a detailed explanation of why the proposed method significantly improves GCD performance. While the alternative training of model and prompt, which enables more fine-grained augmentation, is acknowledged, the paper does not thoroughly describe how this relates to the GCD problem and why it leads to better performance.

- The reasoning behind why SPTNet outperforms Global Prompt in GCD is supported only by experimental results and not by direct consideration of object parts, with insufficient evidence contrary to Vaze et al. (2022)'s findings.

- The paper needs analysis of how alternative training induces changes and what benefits it has over end-to-end or completely separate two-stage learning strategies in terms of Expectation-Maximization (EM) learning aspects.

**Questions:**

The paper could strengthen its reasoning and analysis linking the simple strategy proposed to the task of GCD. While it demonstrates a substantial performance increase with a straightforward approach, there is a lack of analysis or reasoning provided in the paper, making it difficult to directly correlate the performance improvements with their causes.

---

> ### Author Response · Authors · 2023-11-21
> **Response to Reviewer #Pson**
>
> > Lack of a detailed explanation of why the proposed method significantly improves GCD performance…The paper does not thoroughly describe how this relates to the GCD problem and why it leads to better performance.
>
> We thank the reviewer for the comment. Indeed our design is specialized for GCD. Please refer to General Response - Q{1,2,3} for further discussion, more qualitative analysis, and more quantitative study to demonstrate the specialty of our method for GCD.
>
> > The reasoning behind why SPTNet outperforms Global Prompt in GCD is supported only by experimental results and not by direct consideration of object parts, with insufficient evidence contrary to Vaze et al. (2022)'s findings.
>
> Thanks for the suggestion. We have followed the suggestion to provide direct consideration of objects parts by qualitative analysis. Please refer to General Response - Q2.
>
> > The paper needs analysis of how alternative training induces changes and what benefits it has over end-to-end or completely separate two-stage learning strategies in terms of Expectation-Maximization (EM) learning aspects.
>
> Thanks for the suggestion. We have followed the suggestion to provide additional quantitative analysis in General Response - Q3 to further demonstrate the effectiveness of our EM-inspired alternative learning design.
> Additionally, in Appendix E and F, we qualitatively demonstrate the impact of alternative training on both data parameters and model parameters. Our results reveal that compared to end-to-end training, alternative training effectively prevents prompt to be zero and thus ensures the diversity of augmentation for contrastive learning. We also provide theoretical analysis from the perspective of EM for the reason using alternative training in Appendix G, which shows that end-to-end training is not equivalent to alternate training.
> Meanwhile, our alternative training design separates the training model parameters and data parameters into different training steps, resulting in reduced training time and a manageable number of learnable parameters in each training step. Consequently, this approach alleviates the training difficulty.
>
>
> > The paper could strengthen its reasoning and analysis linking the simple strategy proposed to the task of GCD. While it demonstrates a substantial performance increase with a straightforward approach, there is a lack of analysis or reasoning provided in the paper, making it difficult to directly correlate the performance improvements with their causes.
>
> We appreciate the suggestion. We follow this suggestion to include more reasoning and analysis, both empirically and theoretically for GCD. Particularly, we provide more quantitative analysis on the alternative training in General Response - Q3, qualitative analysis on the spatial prompt learning in Appendix E, qualitative analysis on the learned representation in Appendix F, and theoretical analysis in Appendix G. All these consistently demonstrate the effectiveness of our design for GCD. Additionally, the results and discussion in Table S2, S3, and S4 for addressing the comments of Reviewer E8qj further strengthens the reasoning of our design and links to GCD.

---

### Official Review · Reviewer_bfKG · 2023-11-09

**Soundness:** 3 good
**Presentation:** 4 excellent
**Contribution:** 3 good
**Rating:** 6
**Confidence:** 2

**Summary:**

This paper approaches Generalized Category Discovery (GCD) from an alternate perspective, which optimizes data and model parameters by prompt learning and finetuning, respectively. To this end, they propose a visual prompt learning method that learns data representation to better focus on object parts for generalizability. They achieve further performance improvement compared with the previous arts and investigate their approach with sufficient experimental analyses.

**Strengths:**

- The paper is well-organized to present their approach. They first tackle the previous GCD methods and redefine the problem with their own perspective. Then, they show their proposed methods based on their objective. It is easy to follow their objective that they propose spatial prompt tuning for better generalization on both seen and unseen classes.
- The authors demonstrate the effectiveness of SPNet in their framework with sufficient experimental analysis. Their in-depth analysis shows that their proposed method clearly contributes to performance improvement.

**Weaknesses:**

- Although the authors explain the necessity of an alternative training strategy by referring to the EM algorithm, this reviewer did not reach the reasoning behind this explanation. This reviewer can agree that the authors demonstrate this training strategy empirically. It does not seem to be a specialized method for GCD. This reviewer recommends explaining in more detailed reasoning to choose this strategy if the authors have more reasons than the only empirical observation.
- Even though the authors investigate the alternative training strategy by ablation study, this reviewer suggests presenting the visualization of representation during training at each switch to show the representation is enhanced as the objective of their approach.
- As far as this reviewer’s understanding, their framework can be utilized for zero-shot learning tasks such as open-set recognition and open-vocabulary semantic segmentation, which evaluate the model on both seen and unseen classes, not only GCD. This reviewer agrees that their results show efficiency and efficacy. This reviewer believes that the experiment results on the closely related task strengthen their study.

**Questions:**

The questions are naturally raised in the weaknesses section.

---

> ### Author Response · Authors · 2023-11-21
> **Response to Reviewer #bfKG**
>
> >  It does not seem to be a specialized method for GCD. This reviewer recommends explaining in more detailed reasoning to choose this strategy
>
>
> We thank the reviewer for the comment. Indeed our design is specialized for GCD. Please refer to General Response - Q{1,2,3} for further discussion, more qualitative analysis, and more quantitative study to demonstrate the specialty of our method for GCD
>
>
> > Presenting the visualization of representation during training at each switch to show the representation is enhanced as the objective of their approach.
>
> Thanks for the suggestion. We follow the suggestion to conduct visualization to verify the effectiveness of our alternate training strategy for representation learning on CIFAR-10 dataset (Please refer to the newly added Appendix F). Specifically, we present the model representations at the $20^{th}$, $100^{th}$, $200^{th}$, and $400^{th}$ epochs before switching . With the increase of  epoch, we observe clearer boundaries between semantic categories and increased compactness within each cluster. This confirms that our alternate training strategy leads to more robust representation.
> In addition, we provide more discussion in General Response -Q1, more qualitative analysis in General Response -Q2, and more quantitative analysis in General Response -Q3.
>
> > Their framework can be utilized for zero-shot learning tasks such as open-set recognition and open-vocabulary semantic segmentation, which evaluate the model on both seen and unseen classes, not only GCD. This reviewer agrees that their results show efficiency and efficacy. This reviewer believes that the experiment results on the closely related task strengthen their study.
>
> We appreciate the reviewer's recognition of the efficiency and efficacy of our framework and the suggestions on potential applications of our framework beyond GCD.
> Indeed, as discussed and analyzed in General Response - Q{1,2,3}, the strengths of our methods are designed on purposely to favor the GCD problem. Other prompt tuning approaches (Jia et al., 2022; Bahng et al., 2022) have been developed to enhance fully supervised learning. However, our patch-based design is to realize the unique ``part transfer’’ insight for GCD. We will consider further exploring the other possible applications of our design in the future work.

---

### Author Response · Authors · 2023-11-21
**General response**

We thank reviewers for their constructive and valuable feedback. We are encouraged that the reviewers find our method to be **“simple and highly effective”** and **“easy to implement”** (Reviewer Pson), as well as our paper to be **“well-organized”** (Reviewer bfKG), **“easy to follow”** (Reviewer bfKG) and with **“high-quality writing”** (Reviewer Pson).
The reviewers also agreed that the experiments were **“sufficient”**, **“thorough”** and **“support most of the claims”** (Reviewer bfKG, Pson and E8qj). We are also pleased that reviewers acknowledge that **“in-depth analysis shows that their proposed method clearly contributes to performance improvement”** (Reviewer bfKG).
We have carefully addressed all concerns raised by the reviewers. First, we provide a **general response** to shared concerns or critical points. We also address the reviewers’ individual concerns after their comments. We will further strengthen the final manuscript based on the final comments from the reviewers.


### **Q1. Specificity of the method to GCD (Reviewers bfKG and Pson)**


A key insight in GCD is that leveraging *object parts* is a very effective mechanism for transferring knowledge between ‘old’ classes and ‘new’ ones (Vaze et al., 2022). The design of SPT aims to bias the attention to fine-grained object regions, as opposed to global prompts (Jia et al., 2022; Bahng et al., 2022). Moreover, we consider SPT as a learned data augmentation for the GCD clustering task. Data augmentations are of central importance in contrastive learning, which is in turn a core component of current GCD methods (Vaze et al., 2022; Cao et al., 2022; Wen et al., 2023; Pu et al., 2023; Zhang et al., 2023).

We suggest that the joint importance of such transfer from ‘old’ to ‘new’ categories, as well as a contrastive learning signal, is unique to GCD. Specifically, it is not present in tasks like fully supervised recognition (where global prompting is often applied) or self-supervised learning (in which contrastive learning is often applied).

We thank the reviewers for raising this point, and we will include this in the paper’s discussion.

### **Q2. Qualitative evidence for suitability of SPTNet for GCD (Reviewers bfKG and Pson)**

To enhance the clarity of Fig. 4(b), we present the attended regions with only the top 10% most attended patches visualized. These are shown in red and demonstrate circumstances in which SPTNet seems to provide qualitative gains over the baseline. Specifically, the application of SPTNet results in more diverse attention maps across different heads (top right subfigure), with a stronger focus on the semantic parts of the foreground object (bottom right subfigure). We suggest that this provides qualitative support for the motivations behind SPTNet, detailed above and in the paper.


### **Q3. Further ablation of the alternate training strategy (Reviewers bfKG, Pson and E8qj)**


We appreciate the suggestions from the reviewers and have included the requested experiments. To evaluate the efficacy of the alternate training strategy, we conducted additional experiments on fine-grained datasets, and summarized the results in Table S1. Our proposed strategy suppasses the alternatives: (3rd row) training prompt parameters until convergence first, followed by training model parameters until convergence; (4th row) training model parameters until convergence first, followed by training prompt parameters until convergence.

We set the frequency of stage switching as $k=1$ when fine-tuning the parameters of the other component. While both Rows 3 and 4 outperform the fine-tuned SimGCD, they are inferior to the proposed alternate training strategy. We suggest that the alternate learning method provides a **better tradeoff between joint parameter training and optimization stability**. While the input parameters remain fixed, the model parameters are updated, and vice versa. We further note that our proposed method optimizes far fewer parameters than fully finetuning the backbone (Row 2), pointing to the efficacy of the method.

Table S1: Evaluation on SSB. Bold values represent the best results.
| Methods                     | #Trainable parameters (#model parameters + #data parameters) | All      | Old      | New      |
|-----------------------------|---------------------------|----------|----------|----------|
| SimGCD                      | **14.1M+0M**                     | 56.1     | 65.5     | 51.5     |
| SimGCD (fine-tuned)         | 91.1M+0M                     | 57.0     | 66.0     | 52.3     |
| SPTNet (prompt first)       | 14.1M+0.1M                      | 58.0     | 66.4     | 51.9     |
| SPTNet (model first)        | 14.1M+0.1M                      | 59.2     | 67.8     | 54.9     |
| SPTNet (alternate training) | 14.1M+0.1M                      | **61.4** | **69.9** | **57.5** |

---

### Meta-Review · Area_Chair_fiDj · 2023-12-08

**Metareview:**

The paper proposes an alternative approach to Generalized Category Discovery (GCD) by optimizing data and model parameters through prompt learning and finetuning, respectively. The authors introduce a visual prompt learning method that learns data representation to better focus on object parts for generalizability. Their approach achieves a significant performance improvement of about 10% over existing methods using fewer parameters. The results obtained are strong and the presentation is clear. I recommend accepting this paper as a Poster Presentation.

**Justification For Why Not Higher Score:**

As brought up by reviewers, some more analysis on the importance of model components and the reason why they work would improve the paper. In its curent form this work perfectly fits the poster format.

**Justification For Why Not Lower Score:**

The proposed method works well and clearly outperforms the previous work.

---

### Decision · Program_Chairs · 2024-01-16

Accept (poster)